# A new statistical approach to improve the satellite based estimation of the radiative forcing by aerosol- cloud interactions

Piyushkumar N Patel[1], Johannes Quaas[2], Raj Kumar[1]

**[1]**Space Applications Centre, ISRO, Ahmedabad, India

[2]Institute for Meteorology, Universität Leipzig, Leipzig, Germany

*Corresponding to*: Piyushkumar N Patel (piyushether@gmail.com)

## Abstract

In a previous, study of *Quaas et al., (2008)* the radiative forcing by anthropogenic aerosol due to aerosol-cloud interactions, $RF_{aci}$, was obtained by a statistical analysis of satellite retrievals using a multilinear regression. Here we employ a new statistical approach to obtain the fitting parameters, determined using a non-linear least square statistical approach, for the relationship between planetary albedo and cloud properties and, further, the relationship between cloud properties and aerosol optical depth. In order to verify the performance, the results from both statistical approaches (previous and present) were compared to the results from radiative transfer simulations over three regions for different seasons. We find that the results of the new statistical approach agree well with the simulated results both over land and ocean. The new statistical approach increases the correlation by 21%-23% and reduce the error, compared to the previous approach.

## 1   Introduction

Aerosols are considered to have a large effect on climate, both through aerosol radiation interactions, and through aerosol-cloud interactions by serving as cloud condensation nuclei (CCN), therefore increasing $N_d$ and thus cloud albedo (*Twomey, 1974*), as well as rapid cloud adjustments (*Boucher et al., 2013*). Much work has been done to quantify the radiative forcing by aerosol-cloud interaction ($RF_{aci}$), yet it remains highly uncertain. The annual radiative forcing from aerosol induced changes in cloud albedo were reported as -0.7 $Wm^{-2}$ with an uncertainty range -1.8 to -0.3 $Wm^{-2}$ (*Boucher et al., 2013*); this effect could offset much of the warming from greenhouse gases (*Huber and Knutti, 2011*), emphasizing the need to understand the effect so that we can better predict the future climate.

In this study, we concentrate on the $RF_{aci}$, the change in cloud albedo with increasing aerosol. An increasing aerosol at constant cloud water content is supposed to decrease droplet size, which in turn increases the cloud albedo due to the increase scattering of the smaller, more numerous cloud droplets. *Feingold et al. (2001, 2003); McComiskey et al., (2009)* proposed a metric to quantify the microphysical component of the cloud albedo effect ($ACI = - d \ln N_d / d \ln \alpha$), where $N_d$ is the cloud droplet number concentration and $\alpha$ in some proxy for the aerosol burden. A variety of proxies has been used to represent the cloud response to the change in aerosol, e, g., cloud optical depth ($\tau_c$), cloud drop number concentration ($N_d$) and cloud droplet effective radius ($r_e$). Similarly, various proxies have been used to represent the total ambient aerosol burden, including aerosol number concentration ($N_a$), aerosol optical depth ($\tau_a$) and aerosol index (AI). An overview about published relationships and their biases

due to mismatches between process- and analysis scales are discussed in *McComiskey and Feingold, (2012)*. Values for ACI metrics from observations often differ significantly from model-based values (*Quaas et al., 2008, 2009; Bellouin et al., 2008; Penner et al., 2011, 2012*). For example, the observational-based values of $RF_{aci}$, often in the range of -0.2 to -0.6 $Wm^{-2}$ (*Quaas et al., 2008; Bellouin et al.,2013*), is tend to be weaker than the modeled values in the range of -0.5 to -1.9 $Wm^{-2}$ (*IPCC, 2007*). The differences in model and observational-based $RF_{aci}$ have to be reconciled. *Penner et al., (2011)* reported that the lower sensitivities of cloud droplet number concentration, when considering aerosol optical depth (AOD) compared to aerosol index as aerosol quantity may lead to a significant underestimation in satellite-based $RF_{aci}$. However, *Quaas et al., (2011)* pointed out the weaknesses in the approach used by *Penner et al., (2011)*. Clearly, further study is needed to reduce the uncertainties in both observational- and model-based estimates of aerosol RFaci and to reconcile the differences.

*Quaas et al., (2008)* derived the anthropogenic aerosol $RF_{aci}$ based on satellite retrievals of aerosol and clouds properties using statistical relationships between cloud properties and anthropogenic aerosols without the use of radiative transfer model. They developed a statistical relationship between planetary albedo and cloud properties using a multilinear fit, and further, the relationships of cloud properties and aerosol optical depth. *Quaas et al. (2008)* suggested that uncertainties in the statistical relationship and fitting parameters introduced uncertainty in the estimate of $RF_{aci}$. Therefore, it is useful to reassess the estimated $RF_{aci}$ by using a new statistical fitting approach. The main objective of this study is to explore the uncertainty in the satellite-based quantification of $RF_{aci}$. This study differs from previous studies by introducing new statistical fitting approach to obtain the fitting parameters for the estimates of $RF_{aci}$, determined using a nonlinear fit between planetary albedo and cloud properties. To verify the present approach, the results from both statistical approaches are compared with the results from a radiative transfer model.

The rapid socio-economic development in the recent past has increased the anthropogenic emissions in the South asian region along with several parts of the world. The South Asian ones are among the potential sources of a variety of aerosol species; both natural and anthropogenic, and extensive investigations are being made in the past years (e.g., *Chin et al., 2000; Di Girolamo et al., 2004; Moorthy et al., 2013*). These densely populated regions with the increasing power demand, fuel consumption and equally diverse geographical features are also vulnerable to the impacts of atmospheric aerosols to the climate (e.g. *Liu et al., 2009*). The complex geography of this region contributes significant amounts of natural aerosols (desert dust, pollen, sea-salt etc) into the atmosphere, which mix with anthropogenic ones, making the aerosol environment one of the most complex in the world (*Moorthy et al., 2015*). The large spatial heterogeneity of the sources coupled with the atmospheric dynamics driven by topography and contrasting monsoons, make South Asia's aerosol very difficult to characterize and to model their implications on radiative and climate forcing. While tropospheric perturbations would produce strong regional signatures, their global impacts still remain marginally above the uncertainty levels (*IPCC, 2013*). In the recent years, several studies are carried out on the aerosol characterization and its direct effect over south Asia, but there have been very few studies reported on the aerosol indirect effect using ground- and satellite-based measurements due to complex aerosol and cloud environments. Therefore, we discuss the $RF_{aci}$ for both anthropogenic and natural fraction of aerosol for a period of six-years (2008-2013) for three different regions of south Asia (**Fig. 1**, Arabian Sea (AS; 63°E-72°E, 7°N-19°N), Bay of Bengal (BOB; 85°E-94°E, 7°N-19°N) and Central India (CI; 75°E-84°E, 20°N-30°N)), having

significantly distinct aerosol environments as a result of variations in aerosol sources and
transport pathways (*Cherian et al., 2013; Das et al., 2015; Tiwari et al., 2015*). Additionally,
we also discuss the uncertainties of the results in the following sections.
**2 Data**
We combine measurements of aerosol, cloud and radiative properties to derive the top-of-
the atmosphere (TOA) $RF_{aci}$ for both anthropogenic and natural aerosols. Data acquired by
MODerate Resolution Imaging Spectroradiometer (MODIS) and Clouds and the Earth's
Radiant Energy System (CERES) mounted on Aqua (*Parkinson, 2003*) and Ozone Monitoring
Instrument (OMI) onboard Aura (*Schoeberl et al., 2006*) are used in this study. We use the
broadband shortwave planetary albedo ($\alpha$) (*Wielicki et al., 1996; Loeb, 2004; Loeb et al., 2007*)
as retrieved by the CERES in combination with cloud properties from the MODIS (*Minnis et*
*al., 2003*) and AOD ($\tau_a$) and fine mode fraction (FMF) as retrieved by the MODIS onboard
Aqua (*Remer et al., 2005*). Albedo and cloud properties are from the CERES Single-Scanner-
Footprint (SSF) Level-2 Edition-3A data set at $20\times20$ km$^2$ horizontal resolution and aerosol
properties (AOD and FMF) at 550nm from the MYD04 level-2 collection-5.1 dataset at $10\times10$
km$^2$ horizontal resolution are used. We used UV-aerosol index (UV-AI; *Torres et al., 1998*)
measured by OMI-AURA (*Levelt et al., 2006*) from the OMAERUVG level-2 version 003
dataset at $0.25°\times0.25°$ grid, which is a gridded dataset containing retrievals from the
OMAERUV (*Torres et al., 2007*) algorithm. The data from CERES and MODIS level-2
products are interpolated to a $0.25°\times0.25°$ regular longitude-latitude grid to separate the aerosol
and cloud properties for anthropogenic and natural aerosols using UV-AI. Daily data, taken at
roughly 13:30 local time, cover the 2008-2013 period.
**3 Methods**
All statistics between aerosol and cloud properties are computed separately for 3 regions
and for each month of data at $0.25°\times0.25°$ grid resolution. To avoid the greater uncertainty that
exists in a clear distinction between aerosols and clouds and accurate retrieval of cloud
properties, only single-layer cloud with liquid water path (LWP) > 20 gm$^{-2}$ are taken into
account. $RF_{aci}$ for anthropogenic and natural aerosols are calculated using the methods outlined
by *Quaas et al., (2008)* with the new statistical approach. As a part of this process, the method
by *Kim et al., (2007)* MODIS-OMI algorithm (MOA) is employed to classify the aerosol types
into one of four types sea-salt, carbonaceous, dust and sulfate using MODIS FMF and OMI
UV-AI data. FMF provides information on the representative size of the aerosol. FMF is close
to 1 for mostly small aerosol particles, which implies an anthropogenic origin and FMF
becomes small for non-anthropogenic aerosol like dust. UV-AI allows to detect the absorption
due to the presence of an aerosol layer by utilizing the sensitivity of absorptive aerosol in UV.
Under most condition, UV-AI is positive for absorbing aerosols and negative for non-absorbing
aerosols. Using these two independent data sets, aerosol can be classified. Details for the
aerosol classification are discussed in *Kim et al., (2007)*. For the purpose of this research, the
combination of dust and sea-salt AOD considered as a natural AOD and an anthropogenic AOD
contains the combination of carbonaceous and sulfate. Further, the $RF_{aci}$ is estimated for both
anthropogenic and natural aerosols.
**3.1 Satellite-based estimate of $RF_{aci}$**
$RF_{aci}$ is a function of the relationship between AOD and $N_d$ in a cloud. $N_d$ is not directly
provided by satellite product and must be computed using cloud optical thickness ($\tau_c$) and
effective droplet radius ($r_e$) for liquid water clouds assuming adiabaticity (*Brenguier et al.,*
*2000; Schüller et al., 2005; Quaas et al., 2006; Bennartz, 2007; Rausch et al., 2010*).

$$N_d = \gamma \tau_c^{1/2} r_e^{-5/2} \tag{1}$$

Where, a constant value of $\gamma = 1.37 \times 10^{-5}$ m$^{-0.5}$ (*Quaas et al., 2006*) is used in this study. A
limitation of this assumption is that it applies rather well for the stratiform clouds in the marine
boundary layer, but less so for convective clouds. A detailed explanation and uncertainty
assessment are described in *Bennartz, (2007) and Rausch et al., (2010)*. Recently, *Bennartz*
*and Rausch, (2017)* show that the uncertainties in the CDNC climatology from 13-years of
AQUA-MODIS observations are in the order of 30% in the stratocumulus regions and 60% to
80% elsewhere and its contribution to the total uncertainty for this study is discussed in the
following section.
*Quaas et al., (2008)* have adopted the *Loeb (2004)* approach for the estimate of planetary
albedo. Albedo ($\alpha$) of a cloud scene can be well described by a sigmoidal fit as

$$\alpha \approx (1-f)[a_1 + a_2 ln\tau_a] + f[a_3 + a_4(f\tau_c)^{a_5}]^{a_6} \tag{2}$$

Where, $a_1$- $a_6$ are fitting parameters obtained by a multilinear regression, where $a_5$ is set as 1
(*Ma et al., 2014*). Dependency of $\tau_a$ is introduced to include the clear part of the scene in the
above equation and f is the cloud fraction. The satellite-based estimate of RF$_{aci}$ for
anthropogenic and natural aerosols can be expressed as

$$\Delta F_{ant/nat}^{RF_{aci}} = f_{liq}.A(f,\tau_c)\frac{1}{3}\frac{d \ln N_d}{d \ln \tau_a}\left[\ln \tau_a - \ln(\tau_a - \tau_a^{ant/nat})\right]S \tag{3}$$

$$where, A(f,\tau_c) = a_4 a_5 a_6 \left[a_3 + a_4(f\tau_c)^{a_5}\right]^{a_6-1}(f\tau_c)^{a_5}$$

$d \ln N_d / d \ln \tau_a$ is the sensitivity of cloud droplet number concentration ($N_d$) to a relative change
in AOD. It is computed as the slope of the linear regression fit between the natural logarithm
of $N_d$ and AOD (*Quaas et al., 2008*). This value is calculated on a month-by-month basis and
is unique to each region studied. $\tau_a$ is the total AOD, whereas, $\tau_a^{ant/nat}$ are the anthropogenic
and natural AOD, respectively, derived from the FMF and UV-AI as estimated above. A(f, $\tau_c$)
is the empirical function relating albedo to f and $\tau_c$. S is the daily mean solar incoming solar
radiation. RF$_{aci}$ is calculated separately for the anthropogenic and natural aerosols for all three
regions for each month.
A goal of the present study is to assess the uncertainty in the satellite-based estimate of the
RF$_{aci}$. For that purpose, we adopted the new statistical nonlinear least square fitting approach
to obtain the six fitting parameters in Eq. (2). Nonlinear least square methods involve an
iterative improvement to parameters values in order to minimize the residual sum of squares
between the observed values and the predicated value of the dependent variables. We used the
Levenberg-Marquardt (L-M) algorithm (*Levenberg, 1944*) in the nonlinear least square
approach to adjust the parameter values in the iterative procedure. This algorithm combines the
Gauss-Newton method and the gradient descent method. In the gradient descent method, the
sum of the squared errors is reduced by updating the parameters in the steepest descent
direction. In the Gauss -Newton method, the sum of the squared errors is reduced by assuming
the least squares function is locally quadratic, and finding the minimum of the quadratic. The
L-M algorithm acts more like a gradient descent method when the parameters are far from the
optimal value and acts more like to Gauss-Newton method when the parameters are close to
their optimal value. More detail of this method is given in the literature (*Levenberg, 1944;*
*Transtrum et al., 2010; Transtrum and Sethna, 2012*). In the present study, instead of
considering $\alpha_5=1$ in the multiple regression, as in *Quaas et al. (2008) and Ma et al., (2014),* we
obtained the values of all six fitting parameters using a nonlinear fitting approach (L-M
algorithm) for each month and region. To get an impression of the performance of our statistical
approach, we correlate α and $RF_{aci}$ at TOA obtained from both statistical fitting methods
(multilinear and nonlinear) vs. α and $RF_{aci}$ simulated by radiative transfer model for all three
regions. The following section describes the detail information about the simulation of α and
$RF_{aci}$ using the radiative transfer model.

## 3.2 Simulation of planetary albedo (α) and $RF_{aci}$

In order to verify both the statistical approaches, we performed a radiative transfer
simulation to obtain α and $RF_{aci}$ for all three regions. Radiative transfer calculations are
performed with the SBDART [Santa Barbara DISORT Atmospheric Radiative Transfer;
*Ricchiazzi et al., 1998*] that is a plane-parallel radiative transfer code based on the DISORT
algorithm for discrete-ordinate-method radiative transfer in multiple scattering and emitting
layered media (*Stamnes et al., 1988*). The discrete ordinate method provides a numerically
stable algorithm to solve the equations of plane-parallel radiative transfer in a vertically
inhomogeneous atmosphere. Simulations are carried out for the solar spectrum (0.2-4.0μm) for
all three regions. Following the study by *Quaas et al., (2008)* study, *Bellouin et al., (2013)*
performed a similar study with MACC reanalysis data, in which RT simulations, using a Monte
-Carlo method, were carried out to obtain the standard deviation for the uncertainty analysis.
However, in the present study, $RF_{aci}$ is simulated using an RT model (SBDART) to validate
the performance of both the statistical approaches used to compute the $RF_{aci}$ using the statistical
relationship between satellite measurements.
In the present study, simulations are carried out to simulate first α and later $RF_{aci}$ for the
given inputs. Here α is evaluated as the ratio of broadband outgoing (or upwelling) shortwave
flux to the incoming (or downwelling) solar flux. Inputs to the model include profiles of
temperature and water vapor which are resolved into 32 layers extending from 1000 to 1 mbar
and come from European Centre for Medium-range Weather Forecast (ECMWF) reanalysis
data. **Table 1** shows the list of input parameters and their source provided to the RT model for
the estimate of $RF_{aci}$. Total columnar amount of atmospheric ozone is provided from OMI-
AURA. Surface albedo is set to 0.15 to represent a typical land cover value for CI and,
predefined option of the ocean surface is used for the oceanic regions (AS and BOB). In the
SBDART model, the cloud parameter inputs are effective droplet radius ($r_e$), liquid water path
(LWP) and the cloud fraction, all of which are taken from MODIS retrievals reported in the
CERES-SSF product. The geometrical thickness of cloud (CGT) is computed as a difference
between cloud top and bottom heights. Cloud top height is taken from CERES-SSF product
and cloud base height is evaluated using the geopotential height profile from ECMWF data.
Only liquid water clouds are considered in the estimation of $RF_{aci}$. The upwelling and
downwelling fluxes are computed individually computed for all three regions at satellite
(MODIS-Aqua as a reference) overpass time.
The local radiative forcing associated with the $RF_{aci}$ is estimated as the difference between
the perturbed and unperturbed radiative fluxes caused by perturbation in $N_d$ due to the addition
of aerosols while keeping the same meteorology. $RF_{aci}$ is diagnosed by making two calls to the
radiative transfer code: the first call used the unperturbed satellite-derived $N_d$ and the second
used perturbed $N_d$ due to anthropogenic and natural aerosols. The numerical evaluation of
radiative flux for the perturbed case starts by determining the finite perturbation of cloud
droplet number concentration ($\Delta N_d$), calculated as follows:

$$\Delta N_d^{ant/nat} = \frac{d \ln N_d}{d \ln \tau_a} \left[ \ln \tau_a - \ln(\tau_a - \tau_a^{ant/nat}) \right] \tag{4}$$

The finite perturbation in $N_d$ are evaluated separately for anthropogenic and natural aerosol to
estimate the radiative flux for the perturbed case. The perturbed value of $N_d'$ ($N_d + \Delta N_d$) is used
to obtain a perturbed value of $r_e$ using Eq. (5) for constant liquid water content because $r_e$ is
used as an input to the radiative transfer code.

$$N_d' = q_l / (\frac{4}{3} \pi r_e^{\,3} \rho_w) \tag{5}$$

Where, $\rho_w$ is the liquid water density, $q_l$ the liquid water content ($q_l$=liquid water path /
geometrical thickness). $RF_{aci}$ is diagnosed as $RF_{unperturbed}$ - $RF_{perturbed}$ radiative fluxes at the top
of the atmosphere, because increased concentrations of aerosol reduce the effective radius of
cloud particles and smaller cloud particles reflect more radiation back to space. The following
section describes the details of regression analysis of $\alpha$ and $RF_{aci}$ performed between values
from statistical-approaches and simulated values.

## 4 Results

### 4.1 Regression analysis

As stated in section 3.1, the satellite-based estimates of $RF_{aci}$ are dependent on the fitting
parameters $\alpha_1$-$\alpha_6$, obtained here from the two different statistical fitting approaches (multilinear
and nonlinear). The parameters obtained from these two approaches are listed in **Table S1** for
all three regions investigated in this study. These parameters vary with months since we
conducted both the fitting approaches for each month, but only the mean seasonal parameters
are shown here. The main differences in fitting parameters from both methods are found in the
values of $\alpha_4$, $\alpha_5$ and $\alpha_6$. The magnitude of the coefficients $\alpha_4$ and $\alpha_6$ is larger in the nonlinear fit
than the multilinear regression fitting, which may reduce the magnitude of the coefficient $\alpha_5$.
To accomplish the objective of this study, we correlate $\alpha$ and $RF_{aci}$ at TOA obtained from both
statistical fitting approaches (multilinear and nonlinear) with estimates obtained from radiative
transfer model for all three regions. **Fig. 2** shows scatter density plots of comparison between
model-simulated albedo and the one computed from satellite measurements at 0.25°×0.25° grid
resolution using both statistical methods for all three regions. This regression analysis suggests
that the albedo fitted by the new statistical approach (nonlinear fit) agrees well with the model-
simulated albedo over both land and ocean. The scatter of the results from the nonlinear fit
around the 1:1 line is much smaller compared to multilinear fit, which is also reflected in the
coefficients of determination ($R^2$) ranging from 0.74 to 0.79. However, a reduction in over and
underestimation at very large and very small albedos, respectively, is found in the nonlinear fit
compared to the multilinear statistical approach. This is also clearly reflected in the values for
the root mean square error (RMSE), which reduces from 0.042-0.065 to 0.010-0.017,
supporting the expectation that the new statistical method is more reliable. Additionally, a
comparison between the planetary albedo computed using both statistical fits and the CERES
retrieved albedo is shown in **Fig. S1** for all three regions. Similar to the results discussed above,
the analysis shows a good agreement between the CERES derived albedo and the one
calculated using the nonlinear fit.

In addition, we performed a comparison of $RF_{aci}$ obtained from satellite measurements using both statistical approaches with the one simulated by SBDART for each season and for each region. **Fig. 3** illustrates the linear regression of $RF_{aci}$ from the two statistical approaches plotted against the one obtained from the radiative transfer model for both anthropogenic and natural aerosols for all seasons and all three regions. The analysis showed good statistical agreement with Pearson's correlation coefficient r=0.82 and 0.75 and RMSE=0.037 Wm$^{-2}$ and 0.042 Wm$^{-2}$ for the anthropogenic and natural fraction of aerosols, respectively. An examination of **Fig. 3** reveals that the nonlinear fitting approach reduces the scatter seen for the multilinear fit and the improvement in correlation with the simulated forcing. The nonlinear fit increases the correlation by 21%-23% and reduce the RMSE by 0.007-0.011 W m$^{-2}$ compared to the multilinear approach. The relative difference between the RT-simulated and the statistically computed $RF_{aci}$ are computed for both the statistical methods. The mean relative difference in $RF_{aci}$ for anthropogenic fraction of AOD is 0.021 W m$^{-2}$ in the nonlinear and 0.033 W m$^{-2}$ in the multilinear statistical approach, whereas, for $RF_{aci}$ of natural fraction of AOD, it is 0.032 W m$^{-2}$ in nonlinear and 0.053 W m$^{-2}$ in multilinear statistical approach. This suggests that the use of the nonlinear fitting approach reduces the uncertainty by 36%-39% compared to the multilinear regression.

**4.2 $RF_{aci}$ and Uncertainties**

Aerosols and clouds vary substantially as a function of time in all regions; thus it is interesting to analyses aerosol-cloud interactions as a function of season. **Fig. 4** shows the seasonal variability of six-year averaged radiative forcing by aerosol-cloud interaction for the three regions as defined above. The maximum anthropogenic $RF_{aci}$ is found over oceanic regions (AS: -0.15Wm$^{-2}$, BOB: -0.16Wm$^{-2}$), instead of regions over land (CI: -0.12 Wm$^{-2}$) with high anthropogenic emissions. This is because maritime clouds are more susceptible to changes in concentration of anthropogenic aerosols (*Quaas et al., 2008*). In contrast, the natural $RF_{aci}$ is generally stronger over land (-0.15 Wm$^{-2}$) than over oceanic regions (AS: -0.098 Wm$^{-2}$, BOB: -0.07Wm$^{-2}$). It is seen that the anthropogenic $RF_{aci}$ is strongest during winter over AS and BOB, with values near -0.19 Wm$^{-2}$ and -0.22Wm$^{-2}$, whereas it is strong (-0.2 Wm$^{-2}$) during pre-monsoon over CI (land). The dominance of natural aerosols in pre-monsoon results a large natural $RF_{aci}$ both over land (-0.15 Wm$^{-2}$) and ocean (-0.098 Wm$^{-2}$ and -0.07 Wm$^{-2}$).

A direct comparison of the satellite to simulations-based $RF_{aci}$, shows a good correlation. However, both satellite estimated and simulated $RF_{aci}$ are subject to errors and it is useful to compute the associated uncertainties in the above results due to various parameters. Uncertainty involves the ones due to satellite retrievals of AOD which can be highly biased in the vicinity of cloud due to swelling (*Koren et al., 2007*), and also due to 3D effects (*Wen et al., 2007*). Since both biases may be particularly high for thick clouds, our estimate of the $RF_{aci}$ could be still be overestimated. The uncertainty in MODIS retrievals of AOD from validation studies (*Levy et al., 2007*) was quantified at $0.03+0.05\tau_a$ over ocean and $0.05+0.15\tau_a$ over land. However, since we use the MODIS-OMI algorithm (*Kim et al., 2007*) to estimate the anthropogenic and natural fraction of AOD, uncertainty in this is given as 1σ standard deviations as per Table S2. From satellite intercomparison, the uncertainty in radiative flux retrievals by CERES is estimated at 5% (*Loeb, 2004*), and uncertainty in cloud optical depth is 21% (*Minnis et al., 2004*). The uncertainties due to sensitivity of $N_d$ to a relative change in AOD ($d \ln N_d / d \ln \tau_a$) contribute most to the total uncertainty. For $N_d$ sensitivities to changes in AOD, standard deviations are derived from minimum and maximum values obtained for

each season. Following the study by *Bellouin et al., (2013)*, the standard deviations are derived
from minimum and maximum values by defining 4-sigma interval, which covers the large
range of sensitivities and spatio-temporal variabilities. To define the standard deviations in
$RF_{aci}$ due to variation in d ln $N_d$ / d ln $\tau_a$, $RF_{aci}$ is recomputed using those standard deviations
of $N_d$ sensitivities to changes in AOD. Table 2 shows the seasonal and regional sensitivities of
d ln $N_d$ / d ln $\tau_a$ along with their statistical standard deviation, which is computed from the
minimum and maximum values for each season. The associated range in $RF_{aci}$ both for
anthropogenic and natural fraction of AOD is also shown in Table 2, where the standard
deviation of $RF_{aci}$ shows the variation due to change in d ln $N_d$ / d ln $\tau_a$, which finally contribute
to the total uncertainty. In addition to this, the computed $RF_{aci}$ in this study is associated with
the statistical fitting approach as described in section 3. As mentioned earlier, two different
statistical fitting methods are used to obtain the regression coefficients for the estimate of $RF_{aci}$.
In the present study, except for the statistical fitting method, all the variables and
methodologies are same for both the statistical approach. Therefore, we used the relative
difference between the RT-simulated and statistically computed $RF_{aci}$ as an uncertainty due to
the choice of the statistical fitting approach for both the statistical fitting methods. As shown
in section 4.1, the mean relative differences for the nonlinear and multilinear approaches are
0.021 W m$^{-2}$ and 0.033 W m$^{-2}$, respectively, in $RF_{aci}$ for anthropogenic fraction, whereas, for
the $RF_{aci}$ of the natural fraction of AOD, these are 0.032 W m$^{-2}$ and 0.053 W m$^{-2}$ for nonlinear
multilinear statistical approaches, respectively. Table 3 lists the uncertainty due to different
parameters involved in the satellite-based estimate of $RF_{aci}$. We quantify the relative error as
the square root of the sum of the squared relative errors for all individual contributions. This
yields an influence of these relative uncertainties in the input quantities on the computed $RF_{aci}$
of ~±0.08Wm$^{-2}$. It should be noted that we refer here to the published quantifiable uncertainties
in the satellite retrievals. Limitation involved in this approach or uncertainties in the satellite
retrievals contribute to the overall uncertainty, which is difficult to quantify.
In addition to above error budget, there are uncertainties involved in the RT simulated $RF_{aci}$
due to various parameters as shown above. In this regard, the surface albedo plays a major role
in the simulation of $RF_{aci}$. In the standard approach, we have considered a surface albedo value
0.15 for land and the predefined option for the ocean surface albedo is used for the oceanic
regions in the present study. To quantify the uncertainties involved due to assumptions about
the surface albedo, we have simulated $RF_{aci}$ with different plausible surface albedo values and
computed statistics as shown in **Table S3(a) and S3(b)**. The statistics shows that the considered
values of surface albedo are suitably representative of the study regions. In addition, RT
simulation have their own limitations and uncertainties e.g. inherent code accuracy,
overestimate in calculated RF due to plane-parallel bias, 3-D radiative transfer effect etc. It
would be useful to explore these issues in the future. However, in the present study, RT
simulation is used to evaluate the results computed with satellite- based measurements. There
is a scope to improve the present study with the upcoming data set retrieved from spaceborne
active remote sensing instruments, with the improved satellite products and with the new
statistical relationship.
**5   Conclusions**
In this study, we employed a new nonlinear statistical fitting approach to develop the
statistical relationship. A satellite-based algorithm is used to quantify the anthropogenic and
natural fraction of aerosol optical depth for the computation of $RF_{aci}$ from satellite retrievals.
In order to verify, α and $RF_{aci}$ estimates using the new statistical approach (nonlinear) along

with the previous statistical approach (multilinear fit), these are compared with the results obtained from radiative transfer simulations. The results show a better agreement between model-based estimates and the one estimated using the nonlinear approach compared to the multilinear approach. The nonlinear approach relatively increases by 21%-23% the correlation coefficient and reduce RMSE by $0.007 Wm^{-2}$ to $0.011 Wm^{-2}$ compared to multilinear approach. The nonlinear fitting approach reduces the relative difference by 36%-39% compare to the multilinear regression method. The $RF_{aci}$ is found to be consistent with the value found by statistical relationship between aerosol and cloud properties from MODIS and CERES, respectively, and radiative transfer calculations. Further studies using the data retrieved from advanced instruments e.g. lidar and radar may be useful to test the assumption made in the present study concerning the proxy of aerosol column, the overestimation of AOD over land and deal with the multi-layer clouds.

## Acknowledgments

The authors gratefully acknowledge the constant encouragement received from the Director, SAC for carrying out the present research work. Valuable suggestions received from Deputy Director, EPSA, and Head, CVD also gratefully acknowledged. CERES SSF data were obtained from the NASA Langley Research Centre Atmospheric Sciences Data Centre (ASDC), and MODIS data used in this study were acquired as part of NASA(tm)s Earth Science Enterprise. The MODIS Science Teams developed the algorithms for the AOD retrievals. The data were processed by the MODIS Adaptive Processing System for the AOD retrievals. The data were processed by the MODIS Adaptive Processing System and the Goddard Distributed Active Archive (DAAC). The Dutch-Finnish built OMI instrument is part of the NASA EOS Aura satellite payload. The OMI project is managed by NIVR and KNMI in Netherlands. The OMI data were also obtained from the DAAC. Reanalysis data are from the European Centre for Medium-Range Weather Forecasts. The authors would like to thank the data distribution centers for their support, Markand Oza for helpful discussion, and family for their continuous motivation. JQ acknowledges funding by the European Research Council (GA no 306284). The authors are grateful to two anonymous reviewers for constructive and useful comments.

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

**Table 1**: The list of parameters and their sources used  as an input to the SDBART model for
the simulation of RF$_{aci}$.

| Input parameters | Source |
| --- | --- |
| Temperature and Water vapor (for 32 layers extending from 1000 to 1 hPa) | ECMWF reanalysis |
| Total Columnar ozone | OMI-AURA |
| Surface Albedo | For land - 0.15 For ocean - default value of "ocean" (given in SBDART) |
| cloud effective droplet radius cloud liquid water path cloud fraction | MODIS retrievals reported in CERES-SSF product |
| geometrical thickness of cloud | Computed from MODIS and ECMWF data |




**Table 2:** Seasonal and regional sensitivities d ln $N_d$/d ln $\tau_a$ of cloud droplet number concentration $N_d$ to changes in aerosol optical depth used in this study. The given standard deviation is derived from minimum and maximum values for a particular season. The associated range in $RF_{aci}$ is also estimated where the standard deviation of $RF_{aci}$ shows the variation due to change in d ln Nd / d ln $\tau_a$.

| | Region | Winter | Pre-Monsoon | Monsoon | Post-Monsoon |
|---|---|---|---|---|---|
| $\dfrac{d \ln N_d}{d \ln \tau}$ | AS | $0.384 \pm 0.146$ | $0.408 \pm 0.189$ | $0.272 \pm 0.131$ | $0.18 \pm 0.102$ |
| | BOB | $0.314 \pm 0.136$ | $0.414 \pm 0.15$ | $0.194 \pm 0.104$ | $0.148 \pm 0.088$ |
| | CI | $0.214 \pm 0.107$ | $0.178 \pm 0.105$ | $0.107 \pm 0.069$ | $0.122 \pm 0.071$ |
| $Rf_{aci}$ for Anthrophonic Fraction | AS | $-0.19 \pm 0.036$ | $-0.14 \pm 0.056$ | $-0.08 \pm 0.036$ | $-0.16 \pm 0.036$ |
| | BOB | $-0.22 \pm 0.062$ | $-0.16 \pm 0.036$ | $-0.07 \pm 0.02$ | $-0.2 \pm 0.036$ |
| | CI | $-0.13 \pm 0.02$ | $-0.2 \pm 0.036$ | $-0.05 \pm 0.034$ | $-0.16 \pm 0.036$ |
| $Rf_{aci}$ for Natural Fraction | AS | $-0.12 \pm 0.036$ | $-0.18 \pm 0.036$ | $-0.03 \pm 0.04$ | $-0.06 \pm 0.036$ |
| | BOB | $-0.08 \pm 0.026$ | $-0.11 \pm 0.026$ | $-0.04 \pm 0.039$ | $-0.06 \pm 0.017$ |
| | CI | $-0.16 \pm 0.027$ | $-0.22 \pm 0.055$ | $-0.1 \pm 0.027$ | $-0.14 \pm 0.036$ |




**Table 3:** Lists the sources of uncertainties and their values involved in the satellite-based
estimate of $RF_{aci}$ in the present study.

| Source of uncertainty | Values |
|---|---|
| Total AOD | $0.03\pm0.05.\tau_a$ over ocean<br>$0.05\pm0.05.\tau_a$ over land |
| MODIS-OMI algorithm<br>(for the estimate of anthropogenic and natural fraction of aerosol) | $1\sigma$ standard deviation as per below table-S2 |
| Flux retrieval from CERES | 5% |
| Cloud optical depth retrieval from CERES | 21% |
| Cloud droplet number concentration | See table-2 |
| Statistical fitting approach | $0.021$ W m$^{-2}$ in nonlinear for anthropogenic<br>$0.032$ W m$^{-2}$ in nonlinear for natural<br>$0.033$ W m$^{-2}$ in multilinear for anthropogenic<br>$0.053$ W m$^{-2}$ in multilinear for natural |








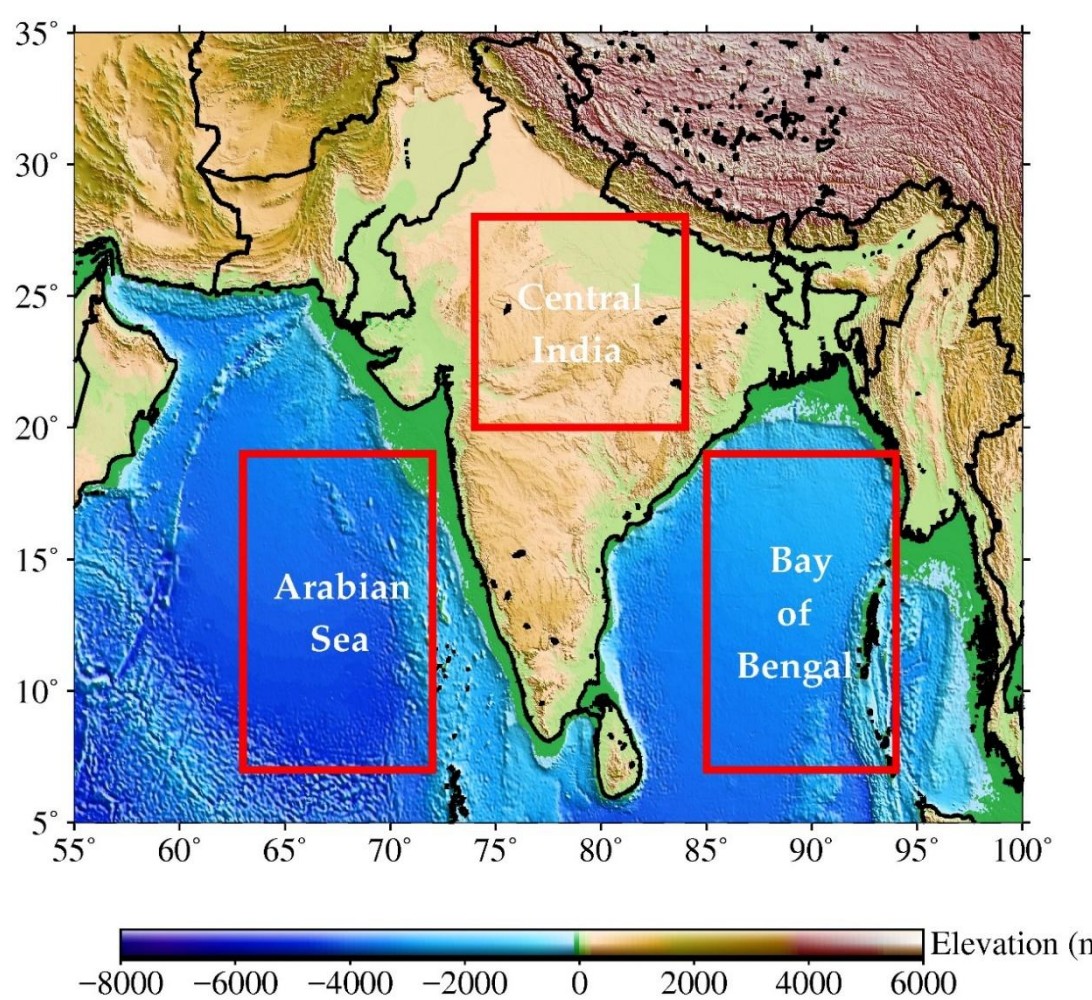


**Figure 1**: Map of India and surroundings showing the study regions. The regions covered by
red box represent the study locations (Arabian Sea, Bay of Bengal and Central India).


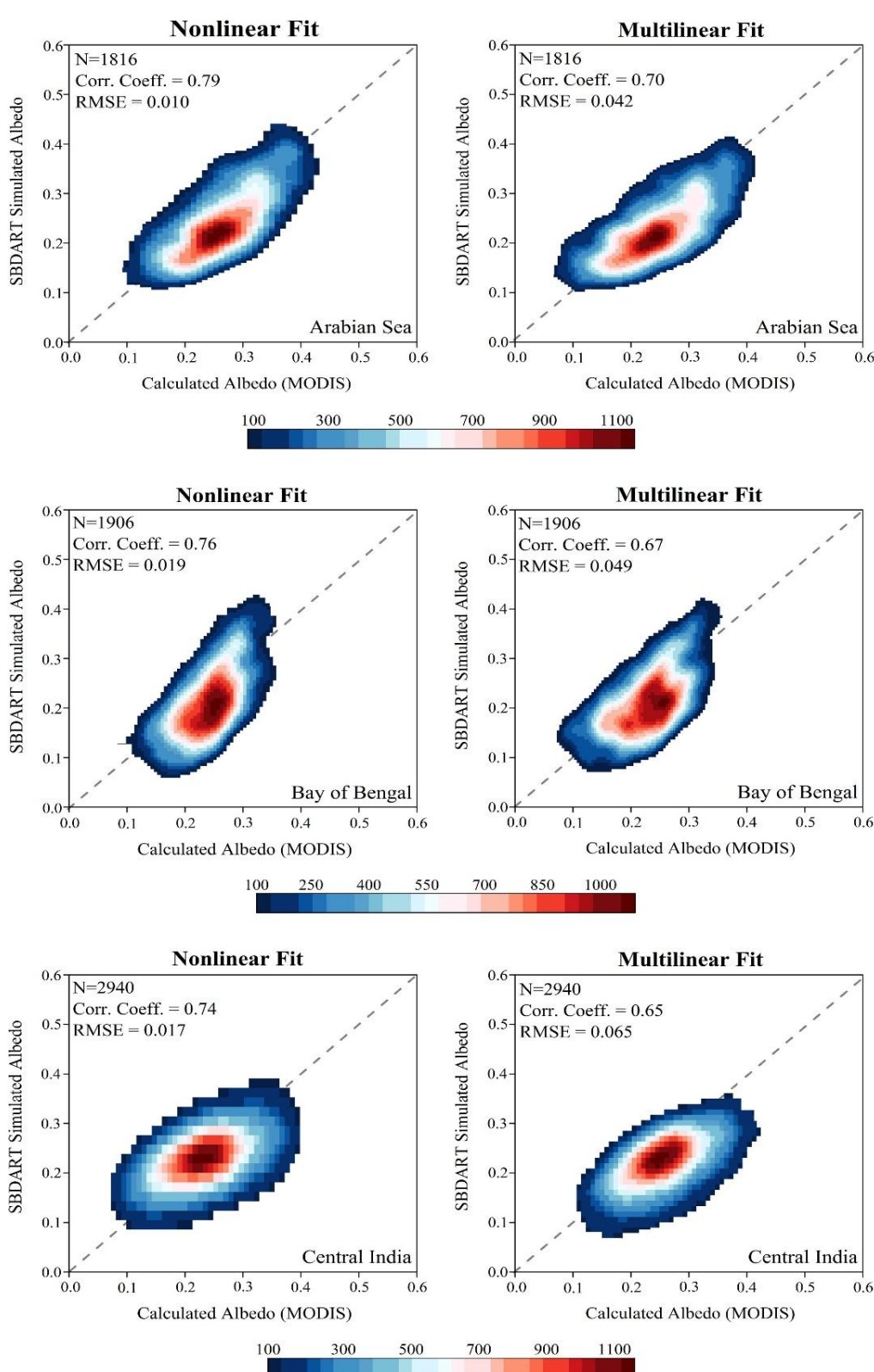


**Figure 2:** Scatter density plots of model-simulated albedo and the one computed using both statistical fitting method (nonlinear and multilinear fit) using satellite measurements for all three regions.





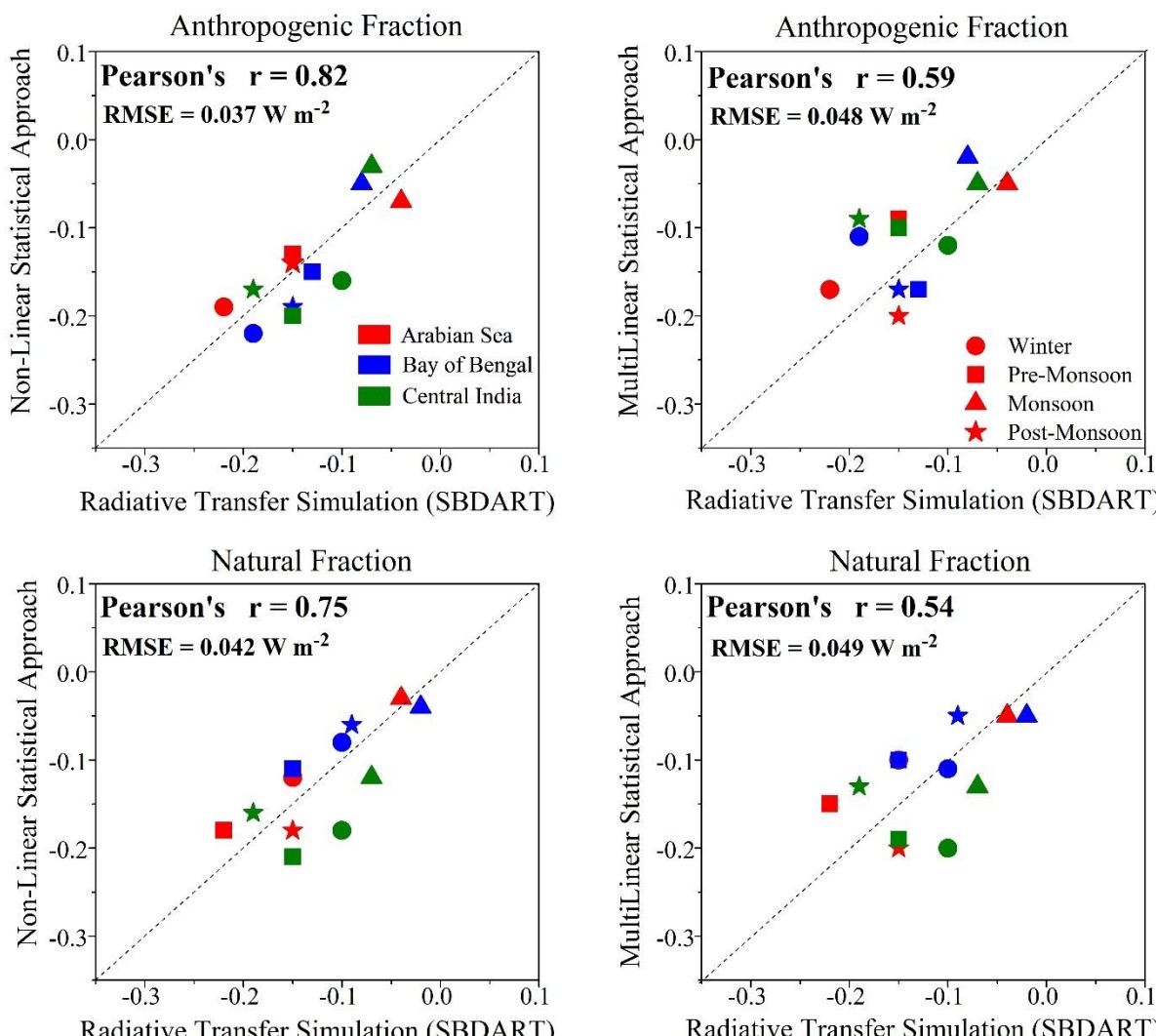

**Figure 3:** Comparison between satellite-based RFaci using both statistical fits and the one simulated by the SBDART model for all three regions and for all seasons. The different color indicates the regions, whereas the different symbols indicates the different seasons. Note that the fit is separately performed for each season and each region.






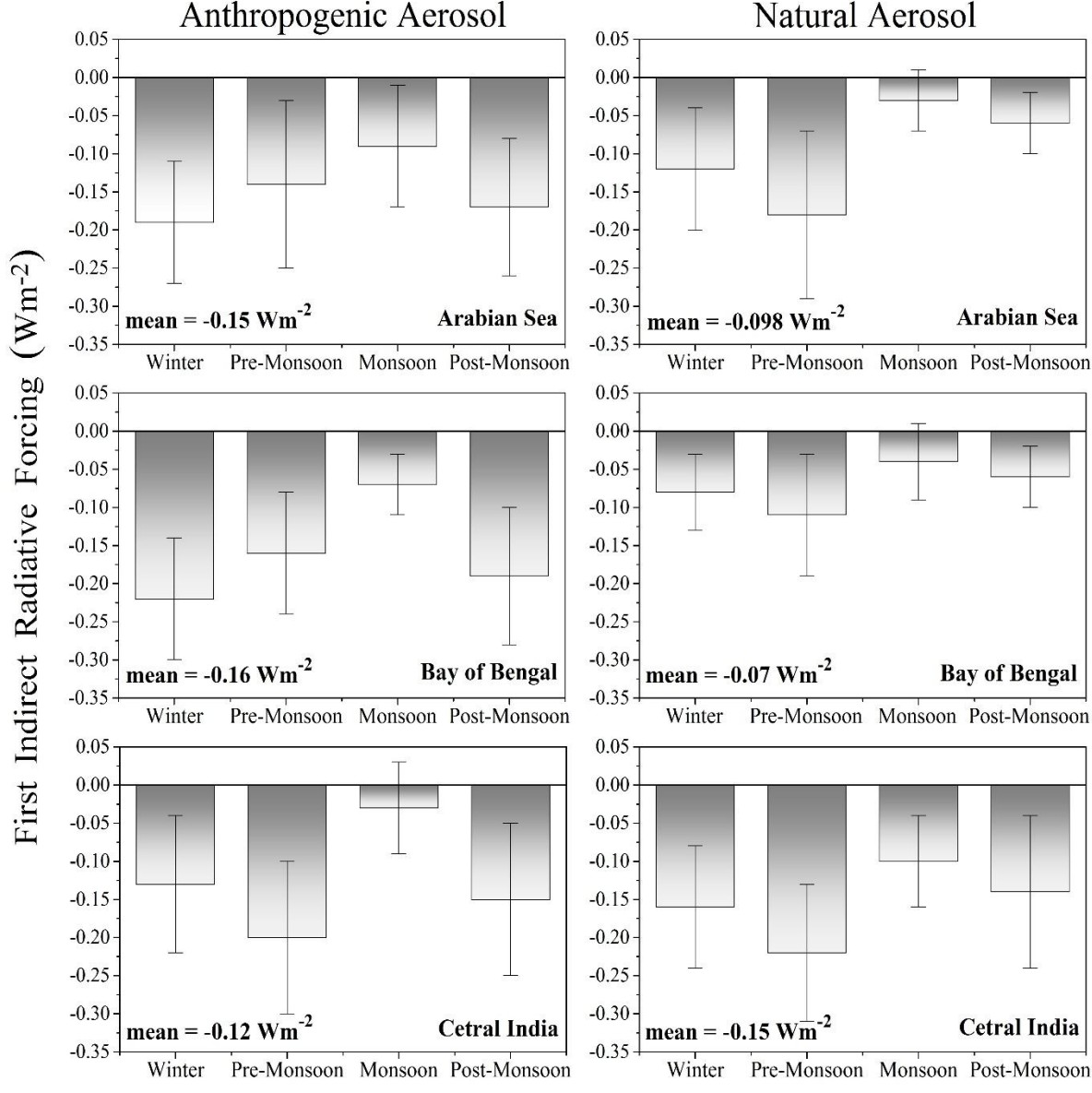

**Figure 4:** Seasonal variability of six-year averaged RFaci obtained using the nonlinear fit for all three regions for both anthropogenic and natural aerosols along with mean values.