# Peer review of "A new statistical approach to improve the satellite based"

_Atmospheric Chemistry and Physics, 2016_

## Referee Comment (RC1) · Anonymous Referee #1 · 22 Sep 2016

The authors have dealt with my suggestions made during preliminary review. Thank you for helping me better understand your study. I had missed that improvement of RMSE was large in my first read through. Overall the paper is well written and referenced.

In regards to the choice of region, I think the discussion does a good job of making the case for South Asia being the most challenging test case for new methods.

Line 133: I think there is a typo on this line. Line 168: How much of an impact does the simplified surface albedo have? I would have thought that the calculation would have some sensitivity to whether it was looking over a forest or farmland. Maybe it averages out, but it seems like it might be enhancing the authors' calculation error. Sea surface

albedo also varies a lot depending on meteorological conditions [Jin et al., 2011].

Line 230: There might be a typo on this line discussing RMSE reduction. Line 262: There is a typo in the last sentence.

I have no other major suggestions relating to this paper and find it acceptable for publication pending minor revisions and grammar corrections.

Jin, Z. H., Y. L. Qiao, Y. J. Wang, Y. H. Fang, and W. N. Yi (2011), A new parameterization of spectral and broadband ocean surface albedo, Optics Express, 19(27), 26429-26443.

---

## Referee Comment (RC2) · Anonymous Referee #3 · 29 Nov 2016

In this work, the authors extend a previous study by Quaas et al. (2008) in several ways. First, an assumption in a fitted, sigmoidal relationship between planetary albedo and aerosol optical depth, cloud fraction, and cloud optical depth is relaxed, and allows the authors to compute the relationship using a non-linear approach. Second, the authors evaluate the relationship between planetary albedo and radiative forcing due to aerosol-cloud interactions ($RF_{ACI}$) derived from satellite data (using their fitting method) compared to offline radiative transfer calculations. Finally, the authors focus their analysis of $RF_{ACI}$ to three small regions and the seasonal cycle of their monsoons. Compared to the offline radiative transfer calculations, using the non-linear approach tends to reduce the root-mean-square error in estimates of planetary albedo and improve correlation relative to the original multilinear approach. Using the non-linear approach, the authors show that in the Bay of Bengal region, natural aerosol produce a smaller $RF_{ACI}$ than anthropogenic aerosol, and that $RF_{ACI}$ is generally much smaller in the monsoon season than at other times of the year.

**0.1 General Comments**

By extending the method of Quaas et al. (2008) to estimate $RF_{ACI}$ and developing and employing a new evaluation scheme for it, this work contributes a useful analysis to the field of aerosol-cloud interactions. Some additional clarification is necessary, though, in order to document how exactly this work complements Quaas et al. (2008) and what advantages it introduces. Furthermore, the manuscript requires extensive copy-editing; as written, some results are hard to understand due to typographical errors and the manuscript is hard to follow at times. The following lists the key issues that must be addressed before publishing:

- I strongly recommend that the authors request copy-editing services from Copernicus to improve the quality of the manuscript. In the **Specific Comments** section I have tried to document typographical and grammatical errors which produce confusion in interpreting the results, but overall there are many such corrections that should be made throughout the document.

- The authors estimate $N_d$ using an adiabatic liquid water cloud assumption. However, this assumption is invalid outside of stratiform clouds in the marine boundary layer, and similar estimates like Bennartz (2007) clearly indicate that this assumption is highly uncertain outside this type of regime. The authors should discuss the limitations of using $N_d$ in their Central India (CI) region, and in seasons dominated by non-stratiform clouds (such as the monsoon one they analyze).

- Several clarifications should be made regarding the non-linear fitting technique.

First, it isn't clear on line 135 why $a_5$ should ever be set to $1$; Quaas et al. (2008) does not seem to make this assumption - contrary to the assertion in lines 146-147 - and if the authors are suggesting this as an alternative formulation of equation (2), then some justification is necessary. For instance, in all except one of the nonlinear fits provided in Table S1, $a_5$ is an order of magnitude smaller than $1$. Second, the authors should clarify what method is used to perform the non-linear fits with a citation if possible, even if it's something standard such as non-linear least squares, for the sake of reproducibility.

- In Section 3.2, the authors present an independent estimate of RF$_{ACI}$ for validation purposes using a radiative transfer code. The authors should include some discussion of how this approach differs from those in the literature, such as Bellouin et al. (2013), and what its limitations are given the dataset and methodology employed. Furthermore, if the use of the radiative transfer code is so readily evaluated in conjunction with satellite data, then what advantage does equation (2) offer in terms of developing constraints for RF$_{ACI}$?

- In equations (3-4) the authors require estimates of $\frac{d \ln N_d}{d \ln \tau_\alpha}$ but do not state where these come from. If they use the regression approach of Quaas et al. (2008), then this should be indicated.

- The discussion of uncertainty in the estimates of RF$_{ACI}$ in Section 4.2 does not seem to follow from the results presented earlier in the manuscript. On lines 258-259 the authors suggest that the nonlinear fitting approach reduces uncertainty by 20%-25%, but it is not clear where this estimate is coming from. The authors' analysis of the reduction in RMSE of planetary albedo compared to the radiative transfer simulations is not a measure of uncertainty, if that's what this statistic refers to. This estimate should be removed, and the authors should instead expand their error-propagation analysis to justify the estimate of $\pm 0.08$ W/m$^2$. For instance, in relation to the previous comment, how does uncertainty in the re-

gional and seasonal estimates of $\frac{d\ln N_d}{d\ln\tau_\alpha}$ influence the estimate of RF$_{\text{ACI}}$?

**0.2  Specific Comments**

- Lines 12-15: This sentence is very awkward and partially repeats itself halfway through.

- Lines 18-20: Sentence needs to clarify what is being compared against with the correlation and error statistics.

- Lines 37-38: Following McComiskey et al. (2009), $\frac{d\ln N_d}{d\ln\tau_\alpha}$ is not computed using partial derivatives and is not calculated with LWP held constant; please remove this statement, or clarify how this relationship differs from the other ACI metrics that could be considered.

- Line 39: Need to define $r_e$ as "droplet effective radius"

- Lines 40-41: Because they are column integrals, metrics like aerosol optical depth do not necessarily represent just the particles impacting clouds - just the total ambient aerosol burden, particularly with respect to larger particles. Please rephrase accordingly.

- Lines 68-74: The first sentence is something of a non-sequitur and could be removed entirely. The second sentence is awkwardly phrased; it would be better to point out that the aerosol mixture in this region is very heterogeneous in time and space with respect to size distribution and chemical composition.

- Lines 82-85: It would be extremely helpful to the reader if you included a figure that outlined where these regions are on a map.

- Lines 91-93: This sentence should be flipped with the following and the beginning of the paragraph re-written to emphasize that your data comes predominantly

from MOIDS and CERES; then you should dive into the details of which data product (and citation) you use for each specific derived quantity.

- Lines 128-131: Pursuant to the general comment about $N_d$, the authors should discuss the limitations of this method for estimating $N_d$

- Line 132: Where does this particular value for $\gamma$ come from?

- Lines 144-152: At a minimum, this paragraph needs additional detail on what nonlinear fitting approach was used (non-linear least squares? some other method?) with a citation if applicable.

- Line 164: Before this sentence, it would be useful if the authors list the variables required to perform their SBDART computations.

- Line 194-185: Please clarify the difference between $\tau_\alpha$ and $\tau_\alpha^{ant/nat}$. Presumably the first is the total AOD and the second is just the anthropogenic/natural contribution to AOD?

- Line 185 and Equation 5: I would recommend writing out explicitly $N_d' = N_d + \Delta N_d$ in both locations.

- Lines 202-203: "Weight" is the wrong word; according to Table S1, it's simply that the magnitude of the coefficients are different.

- Lines 225-227: Rephrase to avoid using terms like "satisfactory results" in preference for neutral language.

- Lines 229-231: The phrasing ". . . decreases RMSE by from 0.007 to 0.011 . . ." is clearly a mistake; please delete whichever word is wrong and be clear about how the RMSE is changing.

Bellouin, N., Quaas, J., Morcrette, J.-J. and Boucher, O.: Estimates of aerosol radiative forcing from the MACC re-analysis, Atmospheric Chemistry and Physics, 13(4), 2045–2062, doi:10.5194/acp-13-2045-2013, 2013.

Bennartz, R.: Global assessment of marine boundary layer cloud droplet number concentration from satellite, Journal of Geophysical Research, 112(D2), doi:10.1029/2006jd007547, 2007.

McComiskey, A., Feingold, G., Frisch, A. S., Turner, D. D., Miller, M. A., Chiu, J. C., Min, Q. and Ogren, J. A.: An assessment of aerosol-cloud interactions in marine stratus clouds based on surface remote sensing, Journal of Geophysical Research, 114(D9), doi:10.1029/2008jd011006, 2009.

Quaas, J., Boucher, O., Bellouin, N. and Kinne, S.: Satellite-based estimate of the direct and indirect aerosol climate forcing, Journal of Geophysical Research: Atmospheres, 113(D5), n/a–n/a, doi:10.1029/2007jd008962, 2008.
* * *

---

## Author Comment (AC1) · 2 Feb 2017

The authors have dealt with my suggestions made during preliminary review. Thank you for helping me better understand your study. I had missed that improvement of RMSE was large in my first read through. Overall the paper is well written and referenced. In regards to the choice of region, I think the discussion does a good job of making the case for South Asia being the most challenging test case for new methods.

Authors: We express our sincere thanks to the anonymous referee for his/her insightful and constructive comments and suggestions on this study. The comment/suggestions were to-the-point and very valuable for us to improve the scientific and technical clarity and quality of the manuscript. In the following, we itemize our point-by-point response

to each of the concerns raised by the referee.

Comment:1 Line 133: I think there is a typo on this line.

Authors: The sentence is rewritten in the revised manuscript.

Comment:2 Line 168: How much of an impact does the simplified surface albedo have? I would have thought that the calculation would have some sensitivity to whether it was looking over a forest or farmland. Maybe it averages out, but it seems like it might be enhancing the authors' calculation error. Sea surface albedo also varies a lot depending on meteorological conditions [Jin et al., 2011].

Author: In the present study, the surface albedo is used to simulate RFaci using the RT model, therefore the simulated values of RFaci are sensitive to the choice of the surface albedo but not the one computed using the statistical relationship between satellite based measurements. To estimate the sensitivity of the simulated RFaci to surface albedo in response to the reviewer's remark, we used different plausible values of surface albedo in sensitivity simulations with radiative transfer model to assess its impact on the simulate the RFaci and to compute the uncertainty statistics. These statistics are now reported in the revised manuscript and presented as supplementary material.

Comment:3 Line 230: There might be a typo on this line discussing RMSE reduction.

Author: The sentence is revised and rewritten in the revised manuscript.

Comment:4 Line 262: There is a typo in the last sentence. I have no other major suggestions relating to this paper and find it acceptable for publication pending minor revisions and grammar corrections.

Author: The sentence is rewritten in the revised manuscript.

Please also note the supplement to this comment:
http://www.atmos-chem-phys-discuss.net/acp-2016-680/acp-2016-680-AC1-

supplement.zip

---

## Author Comment (AC2) · 2 Feb 2017

Response to the Anonymous Referee#3's comments

General Comments:

By extending the method of Quaas et al. (2008) to estimate RFaci and developing and employing a new evaluation scheme for it, this work contributes a useful analysis to the field of aerosol-cloud interactions. Some additional clarification is necessary, though, in order to document how exactly this work complements Quaas et al. (2008) and what advantages it introduces. Furthermore, the manuscript requires extensive copy-editing; as written, some results are hard to understand due to typographical errors

and the manuscript is hard to follow at times.

Authors: We express our sincere thanks to the referee for his/her insightful and constructive comments and suggestions on this study. The comment/suggestions were to-the-point and very valuable for us to improve the scientific and technical clarity and quality of the manuscript. In the following, we itemize our point-by-point response to each of the concerns raised by referee.

Comment:1 I strongly recommend that the authors request copy-editing services from Copernicus to improve the quality of the manuscript. In the Specific Comments section I have tried to document typographical and grammatical errors which produce confusion in interpreting the results, but overall there are many such corrections that should be made throughout the document.

Authors: The given specific comments are seriously considered and the typographical and grammatical errors are corrected throughout the document with the help of an expert to improve the quality of the manuscript.

Comment:2 The authors estimate Nd using an adiabatic liquid water cloud assumption. However, this assumption is invalid outside of stratiform clouds in the marine boundary layer, and similar estimates like Bennartz (2007) clearly indicate that this assumption is highly uncertain outside this type of regime. The authors should discuss the limitations of using Nd in their Central India (CI) region, and in seasons dominated by non-stratiform clouds (such as the monsoon one they analyze).

Authors: The limitation of the computation of Nd from satellite retrievals, and the associated uncertainty along with its contribution to the total uncertainty is discussed in the revised manuscript as suggested by the reviewer. This also takes into account new results on the retrievals of Nd from satellites.

Comment:3 Several clarifications should be made regarding the non-linear fitting technique. First, it isn't clear on line 135 why a5 should ever be set to 1; Quaas et al. (2008)

does not seem to make this assumption - contrary to the assertion in lines 146-147 - and if the authors are suggesting this as an alternative formulation of equation (2), then some justification is necessary. For instance, in all except one of the nonlinear fits provided in Table S1, a5 is an order of magnitude smaller than 1. Second, the authors should clarify what method is used to perform the non-linear fits with a citation if possible, even if it's something standard such as non-linear least squares, for the sake of reproducibility.

Authors: The assumption of a5 set to be 1 was not clearly indicated in Quaas et al., (2008), apologise for this by the second author of our manuscript! - but a subsequent study, Xiaoyan et al., (2014) reassesses the same study by employing updated satellite products, where it is clearly indicated that for this study a5 was set to 1. This second paper is now referenced in the present study for clarity about this point. Second, as per the reviewer's suggestion, more details about the nonlinear least square method are included in the revised manuscript.

Comment:4 In Section 3.2, the authors present an independent estimate of RFaci for validation purposes using a radiative transfer code. The authors should include some discussion of how this approach differs from those in the literature, such as Bellouin et al. (2013), and what its limitations are given the dataset and methodology employed. Furthermore, if the use of the radiative transfer code is so readily evaluated in conjunction with satellite data, then what advantage does equation (2) offer in terms of developing constraints for RFaci?

Authors: As per the reviewer's suggestions the difference between the RT simulation of Bellouin et al., 2013 and in the present study is now explained in detail in the revised manuscript. Following Quaas et al., (2008), Bellouin et al., (2013) performed a similar study (in terms of radiative forcing due to aerosol-cloud interactions) making use of the MACC aerosol reanalysis data and they used a radiative transfer code including a Monte-Carlo method to obtain the standard deviation for the analysis of uncertainty. However, in the present study, RFaci is simulated using a radiative transfer code (SB-

DART) to evaluate the performance of the statistical approach used to compute the RFaci. The use of a simple, analytical expression to assess the aerosol-cloud radiative forcing has the advantage of being very easily understandable, accessible and applicable.

Comment:5 In equations (3-4) the authors require estimates of d ln Nd / d ln $\tau\alpha$ but do not state where these come from. If they use the regression approach of Quaas et al. (2008), then this should be indicated.

Authors: We are thankful to a reviewer to point out the lack of sufficient information. The values of d ln Nd/ d ln $\tau\alpha$ is obtained here are the same as Quaas et al., (2008); derived using a linear regression. This is now clarified in the revised manuscript.

Comment:6 The discussion of uncertainty in the estimates of RFaci in Section 4.2 does not seem to follow from the results presented earlier in the manuscript. On lines 258-259 the authors suggest that the nonlinear fitting approach reduces uncertainty by 20%-25%, but it is not clear where this estimate is coming from. The authors' analysis of the reduction in RMSE of planetary albedo compared to the radiative transfer simulations is not a measure of uncertainty, if that's what this statistic refers to. This estimate should be removed, and the authors should instead expand their error-propagation analysis to justify the estimate of -0.08 W/m2. For instance, in relation to the previous comment, how does uncertainty in the regional and seasonal estimates of d ln Nd / d ln $\tau\alpha$ influence the estimate of RFaci?

Authors: We are thankful to a reviewer for his/her suggestion to expand the error analysis. According to reviewer's suggestion, we introduce the uncertainties involved in the present study due to different parameters considered in this study. The detailed information, including a table about the uncertainty budget is now included in the revised manuscript.

Specific Comments:

Comment:1 Lines 12-15: This sentence is very awkward and partially repeats itself halfway through.

Authors: The sentence is rewritten in the revised manuscript.

Comment:2 Lines 18-20: Sentence needs to clarify what is being compared against with the correlation and error statistics.

Authors: The sentence is rewritten as per the reviewer's suggestion.

Comment:3 Lines 37-38: Following McComiskey et al. (2009), d ln Nd / d ln $\tau\alpha$ is not computed using partial derivatives and is not calculated with LWP held constant; please remove this statement, or clarify how this relationship differs from the other ACI metrics that could be considered.

Authors: We are sorry to introduce the misleading information in the sentence and according to reviewer's suggestions, the sentence is removed in the revised manuscript.

Comment:4 Line 39: Need to define re as "droplet effective radius"

Authors: re defined as "cloud droplet effective radius" in the revised manuscript.

Comment:5 Lines 40-41: Because they are column integrals, metrics like aerosol optical depth do not necessarily represent just the particles impacting clouds - just the total ambient aerosol burden, particularly with respect to larger particles. Please rephrase accordingly.

Authors: The sentence is revised according to reviewer's suggestion in the revised manuscript.

Comment:6 Lines 68-74: The first sentence is something of a non-sequitur and could be removed entirely. The second sentence is awkwardly phrased; it would be better to point out that the aerosol mixture in this region is very heterogeneous in time and space with respect to size distribution and chemical composition.

Authors: The suggested sentence is removed and others are rewritten in the revised manuscript.

Comment:7 Lines 82-85: It would be extremely helpful to the reader if you included a figure that outlined where these regions are on a map.

Authors: As per the reviewer's suggestions, a figure is included in the revised manuscript, which shows the study regions on a map.

Comment:8 Lines 91-93: This sentence should be flipped with the following and the beginning of the paragraph re-written to emphasize that your data comes predominantly from MODIS and CERES; then you should dive into the details of which data product (and citation) you use for each specific derived quantity.

Authors: The sentence is revised as per reviewer's suggestions.

Comment:9 Lines 128-131: Pursuant to the general comment about Nd, the authors should discuss the limitations of this method for estimating Nd.

Authors: The limitation of Nd along with its contribution to the total uncertainty is discussed in the revised manuscript as per suggestion.

Comment:10 Line 132: Where does this particular value for $\gamma$ come from?

Authors: The reference for the value of $\gamma$ is added in the revised manuscript.

Comment:11 Lines 144-152: At a minimum, this paragraph needs additional detail on what nonlinear fitting approach was used (non-linear least squares? some other method?) with a citation if applicable.

Authors: As per reviewer's suggestion, the detail about the nonlinear method is discussed in the revised manuscript.

Comment:12 Line 164: Before this sentence, it would be useful if the authors list the variables required to perform their SBDART computations.

Authors: Following this suggestion, a list of input variables and their sources are tabulated as table-1 in the revised manuscript.

Comment:13 Line 184-185: Please clarify the difference between $\tau\alpha$ and $\tau\alpha$ant/nat. Presumably the first is the total AOD and the second is just the anthropogenic/natural contribution to AOD?

Authors: Yes, it is correct. The difference between both the terms $\tau\alpha$ and $\tau\alpha$ant/nat are clarified in the revised manuscript.

Comment:14 Line 185 and Equation 5: I would recommend writing out explicitly N_dˆ'=N_d+$\Delta$N_d in both locations.

Authors: Terms are modified as per reviewer's suggestions.

Comment:15 Lines 202-203: "Weight" is the wrong word; according to Table S1, it's simply that the magnitude of the coefficients are different.

Authors: The word "weight" is replaced by "magnitude of the coefficient" as per reviewer's suggestion.

Comment:16 Lines 225-227: Rephrase to avoid using terms like "satisfactory results" in preference for neutral language.

Authors: The term "satisfactory results" is replaced in the revised manuscript.

Comment:17 Lines 229-231: The phrasing ". . . decreases RMSE by from 0.007 to 0.011 . . ." is clearly a mistake; please delete whichever word is wrong and be clear about how the RMSE is changing.

Authors: The sentence is revised and rewritten in the revised manuscript.

Please also note the supplement to this comment:
http://www.atmos-chem-phys-discuss.net/acp-2016-680/acp-2016-680-AC2-supplement.zip

---

## Author Response (AR1)

**Response to the Anonymous Referee#1's comments (acp-2016-680-RC1)**

The authors have dealt with my suggestions made during preliminary review. Thank you for helping me better understand your study. I had missed that improvement of RMSE was large in my first read through. Overall the paper is well written and referenced. In regards to the choice of region, I think the discussion does a good job of making the case for South Asia being the most challenging test case for new methods.

**Authors:** We express our sincere thanks to the anonymous referee for his/her insightful and constructive comments and suggestions on this study. The comment/suggestions were to-the-point and very valuable for us to improve the scientific and technical clarity and quality of the manuscript. In the following, we itemize our point-by-point response to each of the concerns raised by the referee.

**Comment:1** Line 133: I think there is a typo on this line.

**Authors:** The sentence is rewritten in the revised manuscript.

**Authors change:**

**Line 144:** *Quaas et al., (2008)* have adopted the *Loeb (2004)* approach for the estimate of planetary albedo.

**Comment:2** Line 168: How much of an impact does the simplified surface albedo have? I would have thought that the calculation would have some sensitivity to whether it was looking over a forest or farmland. Maybe it averages out, but it seems like it might be enhancing the authors' calculation error. Sea surface albedo also varies a lot depending on meteorological conditions [Jin et al., 2011].

**Author:** In the present study, the surface albedo is used to simulate RFaci using the RT model, therefore the simulated values of $RF_{aci}$ are sensitive to the choice of the surface albedo but not the one computed using the statistical relationship between satellite based measurements. To estimate the sensitivity of the simulated $RF_{aci}$ to surface albedo in response to the reviewer's remark, we used different plausible values of surface albedo in sensitivity simulations with radiative transfer model to assess its impact on the simulate the $RF_{aci}$ and to compute the uncertainty statistics. These statistics are now reported in the revised manuscript and presented as supplementary material.

**Authors change:**

**Line 330-340:** In addition to above error budget, there are uncertainties involved in the RT simulated $RF_{aci}$ due to various parameters as shown above. In this regard, the surface albedo plays a major role in the simulation of $RF_{aci}$. In the standard approach, we have considered a surface albedo value 0.15 for land and the predefined option for the ocean surface albedo is used for the oceanic regions in the present study. To quantify the uncertainties involved due to assumptions about the surface albedo, we have simulated $RF_{aci}$ with different plausible surface albedo values and computed statistics as shown in **Table S3(a) and S3(b)**. The statistics shows that the considered values of surface albedo are suitably representative of the study regions. In addition, RT simulation have their own limitations and uncertainties e.g. inherent code accuracy, overestimate in calculated RF due to plane-parallel bias, 3-D radiative transfer effect etc. It would be useful to explore these issues in the future.

**Comment:3** Line 230: There might be a typo on this line discussing RMSE reduction.

**Author:** The sentence is revised and rewritten in the revised manuscript.

**Authors change:**

**Line 265-266:** The nonlinear fit increases the correlation by 21%-23% and reduce the RMSE by 0.007-0.011 W m$^{-2}$ compared to the multilinear approach.

**Comment:4** Line 262: There is a typo in the last sentence. I have no other major suggestions relating to this paper and find it acceptable for publication pending minor revisions and grammar corrections.

**Author:** The sentence is rewritten in the revised manuscript.

**Authors change:**

**Line 327-328:** Limitation involved in this approach or uncertainties in the satellite retrievals contribute to the overall uncertainty, which is difficult to quantify.

**Response to the Anonymous Referee#3's comments (acp-2016-680-RC2)**

**General Comments:**

By extending the method of Quaas et al. (2008) to estimate $RF_{aci}$ and developing and employing a new evaluation scheme for it, this work contributes a useful analysis to the field of aerosol-cloud interactions. Some additional clarification is necessary, though, in order to document how exactly this work complements Quaas et al. (2008) and what advantages it introduces. Furthermore, the manuscript requires extensive copy- editing; as written, some results are hard to understand due to typographical errors and the manuscript is hard to follow at times.

**Authors:** We express our sincere thanks to the referee for his/her insightful and constructive comments and suggestions on this study. The comment/suggestions were to-the-point and very valuable for us to improve the scientific and technical clarity and quality of the manuscript. In the following, we itemize our point-by-point response to each of the concerns raised by referee.

**Comment:1** I strongly recommend that the authors request copy-editing services from Copernicus to improve the quality of the manuscript. In the Specific Comments section I have tried to document typographical and grammatical errors which produce confusion in interpreting the results, but overall there are many such corrections that should be made throughout the document.

**Authors:** The given specific comments are seriously considered and the typographical and grammatical errors are corrected throughout the document with the help of an expert to improve the quality of the manuscript. The suggested specific comments are seriously responded point-by-point in the revised manuscript.

**Comment:2** The authors estimate $N_d$ using an adiabatic liquid water cloud assumption. However, this assumption is invalid outside of stratiform clouds in the marine boundary layer, and similar estimates like Bennartz (2007) clearly indicate that this assumption is highly uncertain outside this type of regime. The authors should discuss the limitations of using $N_d$ in their Central India (CI) region, and in seasons dominated by non-stratiform clouds (such as the monsoon one they analyze).

**Authors:** The limitation of the computation of $N_d$ from satellite retrievals, and the associated uncertainty along with its contribution to the total uncertainty is discussed in the revised manuscript as suggested by the reviewer. This also takes into account new results on the retrievals of $N_d$ from satellites.

**Authors change:**

**Lines 137-144:** A limitation of this assumption is that it applies rather well for the stratiform clouds in the marine boundary layer, but less so for convective clouds. A detailed explanation and uncertainty assessment are described in *Bennartz, (2007) and Rausch et al., (2010)*. Recently, *Bennartz and Rausch, (2017)* show that the uncertainties in the CDNC climatology from 13-years of AQUA-MODIS observations are in the order of 30% in the stratocumulus regions and 60% to 80% elsewhere and its contribution to the total uncertainty for this study is discussed in the following section.

**Comment:3** Several clarifications should be made regarding the non-linear fitting technique. First, it isn't clear on line 135 why $\alpha_5$ should ever be set to 1; Quaas et al. (2008) does not seem to make this assumption - contrary to the assertion in lines 146-147 - and if the authors are suggesting this as an alternative formulation of equation (2), then some justification is necessary. For instance, in all except one of the nonlinear fits provided in Table S1, $\alpha_5$ is an order of magnitude smaller than 1. Second, the authors should clarify what method is used to perform the non-linear fits with a citation if possible, even if it's something standard such as non-linear least squares, for the sake of reproducibility.

**Authors:** The assumption of $\alpha_5$ set to be 1 was not clearly indicated in Quaas et al., (2008), apologies for this by the authors of our manuscript! - but a subsequent study, Ma et al., (2014) reassesses the same study by employing updated satellite products, where it is clearly indicated that for this study $\alpha_5$ was set to 1. This second paper is now referenced in the present study for clarity about this point. Second, as per the reviewer's suggestion, more details about the nonlinear least square method are included in the revised manuscript.

**Authors change:**

**Lines 175-177:** In the present study, instead of considering $\alpha_5=1$ in the multiple regression, as in *Quaas et al. (2008) and Ma et al., (2014),* we obtained the values of all six fitting parameters using a nonlinear fitting approach (L-M algorithm) for each month and region.

**Lines 163-175:** For that purpose, we adopted the new statistical nonlinear least square fitting approach to obtain the six fitting parameters in Eq. (3). Nonlinear least square methods involve an iterative improvement to parameters values in order to minimize the residual sum of squares between the observed values and the predicated value of the dependent variables. We used the Levenberg-Marquardt (L-M) algorithm (*Levenberg, 1944*) in the nonlinear least square approach to adjust the parameter values in the iterative procedure. This algorithm combines the Gauss-Newton method and the gradient descent method. In the gradient descent method, the sum of the squared errors is reduced by updating the parameters in the steepest descent direction. In the Gauss-Newton method, the sum of the squared errors is reduced by assuming the least squares function is locally quadratic, and finding the minimum of the quadratic. The L-M algorithm acts more like a gradient descent method when the parameters are far from the optimal value and acts more like to Gauss-Newton method when the parameters are close to their optimal value. More detail of this method is given in the literature (*Levenberg, 1944; Transtrum et al., 2010; Transtrum and Sethna, 2012*).

**Comment:4** In Section 3.2, the authors present an independent estimate of $RF_{aci}$ for validation purposes using a radiative transfer code. The authors should include some discussion of how this approach differs from those in the literature, such as Bellouin et al. (2013), and what its limitations are given the dataset and methodology employed. Furthermore, if the use of the radiative transfer code is so readily evaluated in conjunction with satellite data, then what advantage does equation (2) offer in terms of developing constraints for $RF_{aci}$?

**Authors:** As per the reviewer's suggestions the difference between the RT simulation of Bellouin et al., 2013 and in the present study is now explained in detail in the revised manuscript.

Following Quaas et al., (2008), Bellouin et al., (2013) performed a similar study (in terms of radiative forcing due to aerosol-cloud interactions) making use of the MACC aerosol reanalysis data and they used a radiative transfer code including a Monte-Carlo method to obtain the standard deviation for the analysis of uncertainty. However, in the present study, $RF_{aci}$ is simulated using a radiative transfer code (SBDART) to evaluate the performance of the statistical approach used to compute the $RF_{aci}$. The use of a simple, analytical expression to assess the aerosol-cloud radiative forcing has the advantage of being very easily understandable, accessible and applicable.

**Authors change:**

**Lines 190-195:** Following the study by *Quaas et al., (2008)* study, *Bellouin et al., (2013)* performed a similar study with MACC reanalysis data, in which RT simulations, using a Monte - Carlo method, were carried out to obtain the standard deviation for the uncertainty analysis. However, in the present study, $RF_{aci}$ is simulated using an RT model (SBDART) to validate the performance of both the statistical approaches used to compute the $RF_{aci}$ using the statistical relationship between satellite measurements.

**Comment:5** In equations (3-4) the authors require estimates of $d\ ln\ N_d\ /\ d\ ln\ \tau_\alpha$ but do not state where these come from. If they use the regression approach of Quaas et al. (2008), then this should be indicated.

**Authors:** We are thankful to reviewer to point out the lack of sufficient information. The values of $d\ ln\ N_d/\ d\ ln\ \tau_\alpha$ is obtained here are the same as Quaas et al., (2008); derived using a linear regression. This is now clarified in the revised manuscript.

**Authors change:**

**Lines 151-154:** $d\ ln\ N_d\ /\ d\ ln\ \tau_\alpha$ is the sensitivity of cloud droplet number concentration ($N_d$) to a relative change in AOD. It is computed as the slope of the linear regression fit between the natural logarithm of $N_d$ and AOD (*Quaas et al., 2008*). This value is calculated on a month-by-month basis and is unique to each region studied.

**Comment:6** The discussion of uncertainty in the estimates of $RF_{aci}$ in Section 4.2 does not seem to follow from the results presented earlier in the manuscript. On lines 258-259 the authors suggest that the nonlinear fitting approach reduces uncertainty by 20%-25%, but it is not clear where this estimate is coming from. The authors' analysis of the reduction in RMSE of planetary albedo compared to the radiative transfer simulations is not a measure of uncertainty, if that's what this statistic refers to. This estimate should be removed, and the authors should instead expand their error-propagation analysis to justify the estimate of -0.08 W/m². For instance, in relation to the previous comment, how does uncertainty in the regional and seasonal estimates **of $d\ ln\ N_d\ /\ d\ ln\ \tau_\alpha$** influence the estimate of $RF_{aci}$?

**Authors:** We are thankful to reviewer for his/her suggestion to expand the error analysis. According to reviewer's suggestion, we introduce the uncertainties involved in the present study due to different parameters considered in this study. The detailed information, including a table about the uncertainty budget is now included in the revised manuscript.

The reduction in RMSE of planetary albedo compared to RT simulations is not a measure of uncertainty in the present study. It is the reduction in RMSE of $RF_{aci}$ due to nonlinear least square approach compared to multilinear regression. Additionally we incorporated the mean relative difference in $RF_{aci}$ due to both statistical approaches.

**Authors change:**
**Lines 266-272:** The relative difference between the RT-simulated and the statistically computed $RF_{aci}$ are computed for both the statistical methods. The mean relative difference in $RF_{aci}$ for anthropogenic fraction of AOD is 0.021 W m$^{-2}$ in the nonlinear and 0.033 W m$^{-2}$ in the multilinear statistical approach, whereas, for $RF_{aci}$ of natural fraction of AOD, it is 0.032 W m$^{-2}$ in nonlinear and 0.053 W m$^{-2}$ in multilinear statistical approach. This suggests that the use of the nonlinear fitting approach reduces the uncertainty by 36%-39% compared to the multilinear regression.

**Lines 298-309:** The uncertainties due to sensitivity of $N_d$ to a relative change in AOD (d ln $N_d$ / d ln $\tau_a$) contribute most to the total uncertainty. For $N_d$ sensitivities to changes in AOD, standard deviations are derived from minimum and maximum values obtained for each season. Following the study by *Bellouin et al., (2013)*, the standard deviations are derived from minimum and maximum values by defining 4-sigma interval, which covers the large range of sensitivities and spatio-temporal variabilities. To define the standard deviations in $RF_{aci}$ due to variation in d ln $N_d$ / d ln $\tau_a$, $RF_{aci}$ is recomputed using those standard deviations of $N_d$ sensitivities to changes in AOD. Table 2 shows the seasonal and regional sensitivities of d ln $N_d$ / d ln $\tau_a$ along with their statistical standard deviation, which is computed from the minimum and maximum values for each season. The associated range in $RF_{aci}$ both for anthropogenic and natural fraction of AOD is also shown in Table 2, where the standard deviation of $RF_{aci}$ shows the variation due to change in d ln $N_d$ / d ln $\tau_a$, which finally contribute to the total uncertainty.

**Lines 314-324:** In the present study, except for the statistical fitting method, all the variables and methodologies are same for both the statistical approach. Therefore, we used the relative difference between the RT-simulated and statistically computed $RF_{aci}$ as an uncertainty due to the choice of the statistical fitting approach for both the statistical fitting methods. As shown in section 4.1, the mean relative differences for the nonlinear and multilinear approaches are 0.021 W m$^{-2}$ and 0.033 W m$^{-2}$, respectively, in $RF_{aci}$ for anthropogenic fraction, whereas, for the $RF_{aci}$ of the natural fraction of AOD, these are 0.032 W m$^{-2}$ and 0.053 W m$^{-2}$ for nonlinear multilinear statistical approaches, respectively. Table 3 lists the uncertainty due to different parameters involved in the satellite-based estimate of $RF_{aci}$. We quantify the relative error as the square root of the sum of the squared relative errors for all individual contributions. This yields an influence of these relative uncertainties in the input quantities on the computed $RF_{aci}$ of ~$\pm$0.08Wm$^{-2}$.

**_Specific Comments:_**

**Comment:1** Lines 12-15: This sentence is very awkward and partially repeats itself halfway through.

**Authors:** The sentence is rewritten in the revised manuscript.

**Authors change:**

**Lines 11-14:** Here we employ a new statistical approach to obtain the fitting parameters, determined using a non-linear least square statistical approach, for the relationship between planetary albedo and cloud properties and, further, the relationship between cloud properties and aerosol optical depth.

**Comment:2** Lines 18-20: Sentence needs to clarify what is being compared against with the correlation and error statistics.

**Authors:** The sentence is rewritten as per the reviewer's suggestion.

**Authors change**:

**Lines 14-20:** In order to verify the performance, the results from both statistical approaches (previous and present) were compared to the results from radiative transfer simulations over three regions for different seasons. We find that the results of the new statistical approach agree well with the simulated results both over land and ocean. The new statistical approach increases the correlation by 21%-23% and reduce the error, compared to the previous approach.

**Comment:3** Lines 37-38: Following McComiskey et al. (2009), $d\,ln\,N_d\,/\,d\,ln\,\tau_\alpha$ is not computed using partial derivatives and is not calculated with LWP held constant; please remove this statement, or clarify how this relationship differs from the other ACI metrics that could be considered.

**Authors:** We are sorry to introduce the misleading information in the sentence and according to reviewer's suggestions, the sentence is removed in the revised manuscript.

**Authors change:**

**Lines 34-39:** *Feingold et al. (2001, 2003); McComiskey et al., (2009)* proposed a metric to quantify the microphysical component of the cloud albedo effect ($ACI = -d\,\ln N_d/d\ln\alpha$), where $N_d$ is the cloud droplet number concentration and $\alpha$ in some proxy for the aerosol burden. A variety of proxies has been used to represent the cloud response to the change in aerosol, e, g., cloud optical depth ($\tau_c$), cloud drop number concentration ($N_d$) and cloud droplet effective radius ($r_e$).

**Comment:4** Line 39: Need to define re as "droplet effective radius"

**Authors:** $r_e$ defined as "cloud droplet effective radius" in the revised manuscript.

**Authors change:**

**Lines 37-39:** A variety of proxies has been used to represent the cloud response to the change in aerosol, e, g., cloud optical depth ($\tau_c$), cloud drop number concentration ($N_d$) and cloud droplet effective radius ($r_e$).

**Comment:5** Lines 40-41: Because they are column integrals, metrics like aerosol optical depth do not necessarily represent just the particles impacting clouds - just the total ambient aerosol burden, particularly with respect to larger particles. Please rephrase accordingly.

**Authors:** The sentence is revised according to reviewer's suggestion in the revised manuscript.

**Authors change:**

**Lines 39-41:** Similarly, various proxies have been used to represent the total ambient aerosol burden, including aerosol number concentration ($N_a$), aerosol optical depth ($\tau_a$) and aerosol index (AI).

**Comment:6** Lines 68-74: The first sentence is something of a non-sequitur and could be removed entirely. The second sentence is awkwardly phrased; it would be better to point out that the aerosol mixture in this region is very heterogeneous in time and space with respect to size distribution and chemical composition.

**Authors:** The suggested sentence is removed and others are rewritten in the revised manuscript.

**Authors change:**

**Lines 68-76:** The rapid socio-economic development in the recent past has increased the anthropogenic emissions in the South Asian region along with several parts of the world. The South Asian ones are among the potential sources of a variety of aerosol species; both natural and anthropogenic, and extensive investigations are being made in the past years (e.g., *Chin et al., 2000; Di Girolamo et al., 2004; Moorthy et al., 2013*). These densely populated regions with the increasing power demand, fuel consumption and equally diverse geographical features are also vulnerable to the impacts of atmospheric aerosols to the climate (e.g. *Liu et al., 2009*).

**Comment:7** Lines 82-85: It would be extremely helpful to the reader if you included a figure that outlined where these regions are on a map.

**Authors:** As per the reviewer's suggestions, a figure is included in the revised manuscript, which shows the study regions on a map.

**Authors change:**

**Lines 86-91and Figure-1 :** Therefore, we discuss the $RF_{aci}$ for both anthropogenic and natural fraction of aerosol for a period of six-years (2008-2013) for three different regions of south Asia (**Fig. 1**, Arabian Sea (AS; 63°E-72°E, 7°N-19°N), Bay of Bengal (BOB; 85°E-94°E, 7°N-19°N) and Central India (CI; 75°E-84°E, 20°N-30°N)), having significantly distinct aerosol environments as a result of variations in aerosol sources and transport pathways (*Cherian et al., 2013; Das et al., 2015; Tiwari et al., 2015*).

**Comment:8** Lines 91-93: This sentence should be flipped with the following and the beginning of the paragraph re-written to emphasize that your data comes predominantly from MODIS and CERES; then you should dive into the details of which data product (and citation) you use for each specific derived quantity.

**Authors:** The sentence is revised as per reviewer's suggestions.

**Authors change:**

**Lines 95-103:** Data acquired by MODerate Resolution Imaging Spectroradiometer (MODIS) and Clouds and the Earth's Radiant Energy System (CERES) mounted on Aqua (*Parkinson, 2003*) and Ozone Monitoring Instrument (OMI) onboard Aura (*Schoeberl et al., 2006*) are used in this study. We use the broadband shortwave planetary albedo ($\alpha$) (*Wielicki et al., 1996; Loeb, 2004; Loeb et al., 2007*) as retrieved by the CERES in combination with cloud properties from the MODIS (*Minnis et al., 2003*) and AOD ($\tau_a$) and fine mode fraction (FMF) as retrieved by the MODIS onboard Aqua (*Remer et al., 2005*).

**Comment:9** Lines 128-131: Pursuant to the general comment about $N_d$, the authors should discuss the limitations of this method for estimating $N_d$.

**Authors:** The limitation of $N_d$ along with its contribution to the total uncertainty is discussed in the revised manuscript as per suggestion.

**Authors change:**

**Lines 137-143:** A limitation of this assumption is that it applies rather well for the stratiform clouds in the marine boundary layer, but less so for convective clouds. A detailed explanation and uncertainty assessment are described in *Bennartz, (2007) and Rausch et al., (2010)*. Recently, *Bennartz and Rausch, (2017)* show that the uncertainties in the CDNC climatology from 13-years of AQUA-MODIS observations are in the order of 30% in the stratocumulus regions and 60% to 80% elsewhere and its contribution to the total uncertainty for this study is discussed in the following section.

**Comment:10** Line 132: Where does this particular value for γ come from?

**Authors:** The reference for the value of γ is added in the revised manuscript.

**Authors change:**

**Line 137:** Where, a constant value of $\gamma=1.37\times10^{-5}$ m$^{-0.5}$ (*Quaas et al., 2006*) is used in this study.

**Comment:11** Lines 144-152: At a minimum, this paragraph needs additional detail on what nonlinear fitting approach was used (non-linear least squares? some other method?) with a citation if applicable.

**Authors:** As per reviewer's suggestion, the detail about nonlinear method is discussed in the revised manuscript.

**Authors change:**

**Lines 162-175:** For that purpose, we adopted the new statistical nonlinear least square fitting approach to obtain the six fitting parameters in Eq. (3). Nonlinear least square methods involve an iterative improvement to parameters values in order to minimize the residual sum of squares between the observed values and the predicated value of the dependent variables. We used the Levenberg-Marquardt (L-M) algorithm (*Levenberg, 1944*) in the nonlinear least square approach to adjust the parameter values in the iterative procedure. This algorithm combines the Gauss-Newton method and the gradient descent method. In the gradient descent method, the sum of the squared errors is reduced by updating the parameters in the steepest descent direction. In the Gauss-Newton method, the sum of the squared errors is reduced by assuming the least squares function is locally quadratic, and finding the minimum of the quadratic. The L-M algorithm acts more like a gradient descent method when the parameters are far from the optimal value and acts more like to Gauss-Newton method when the parameters are close to their optimal value. More detail of this method is given in the literature (*Levenberg, 1944; Transtrum et al., 2010; Transtrum and Sethna, 2012*).

**Comment:12** Line 164: Before this sentence, it would be useful if the authors list the variables required to perform their SBDART computations.

**Authors:** Following this suggestion, a list of input variables and their sources are tabulated as table-1 in the revised manuscript.
**Authors change:**
**Lines 200-201: Table 1** shows the list of input parameters and their source provided to the RT model for the estimate of $RF_{aci}$.

**Comment:13** Line 184-185: Please clarify the difference between $\boldsymbol{\tau_\alpha}$ and $\tau_\alpha{}^{ant/nat}$. Presumably the first is the total AOD and the second is just the anthropogenic/natural contribution to AOD?
**Authors:** Yes, it is correct. The difference between both the terms $\tau_\alpha$ and $\tau_\alpha{}^{ant/nat}$ are clarified in the revised manuscript.
**Authors change:**
**Lines 154-155:** $\tau_a$ is the total AOD, whereas, $\tau_a^{ant/nat}$ are the anthropogenic and natural AOD, respectively, derived from the FMF and UV-AI as estimated above.

**Comment:14** Line 185 and Equation 5: I would recommend writing out explicitly $N_d' = N_d + \Delta N_d$ in both locations.
**Authors:** Terms are modified as per reviewer's suggestions.
**Authors change:**
**Lines 220-221:** The perturbed value of $N_d'$ ($N_d + \Delta N_d$) is used to obtain a perturbed value of $r_e$ using Eq. (5) for constant liquid water content because $r_e$ is used as an input to the radiative transfer code.
**Equation -5**

$$N_d' = q_l / (\frac{4}{3}\pi r_e{}^3 \rho_w) \qquad (1)$$

**Comment:15** Lines 202-203: "Weight" is the wrong word; according to Table S1, it's simply that the magnitude of the coefficients are different.
**Authors:** The word "weight" is replaced by "magnitude of the coefficient" as per reviewer's suggestion.
**Authors change:**
**Lines 237-239:** The magnitude of the coefficients $a_4$ and $a_6$ is larger in the nonlinear fit than the multilinear regression fitting, which may reduce the magnitude of the coefficient $a_5$.

**Comment:16** Lines 225-227: Rephrase to avoid using terms like "satisfactory results" in preference for neutral language.
**Authors:** The term "satisfactory results" is replaced in the revised manuscript.
**Authors change:**
**Lines 260-261:** The analysis showed good statistical agreement with Pearson's correlation coefficient r=0.82 and 0.75 and RMSE=0.037 $Wm^{-2}$ and 0.042 $Wm^{-2}$ for the anthropogenic and natural fraction of aerosols, respectively.

**Comment:17** Lines 229-231: The phrasing ". . . decreases RMSE by from 0.007 to 0.011 . . ." is clearly a mistake; please delete whichever word is wrong and be clear about how the RMSE is changing.

**Authors:** The sentence is revised and rewritten in the revised manuscript.

**Authors change:**

[revised manuscript text omitted]

| Source of uncertainty | Values |
|---|---|
| Total AOD | $0.03 \pm 0.05.\tau_a$ over ocean
$0.05 \pm 0.05.\tau_a$ over land |
| MODIS-OMI algorithm (for the estimate of anthropogenic and natural fraction of aerosol) | $1\sigma$ standard deviation as per below table-S2 |
| Flux retrieval from CERES | 5% |
| Cloud optical depth retrieval from CERES | 21% |
| Cloud droplet number concentration | See table-2 |
| Statistical fitting approach | $0.021$ W m$^{-2}$ in nonlinear for anthropogenic
$0.032$ W m$^{-2}$ in nonlinear for natural
$0.033$ W m$^{-2}$ in multilinear for anthropogenic
$0.053$ W m$^{-2}$ in multilinear for natural |

[Figure]

**Figure 1**: Map of India and surroundings showing the study regions. The regions covered by red box represent the study locations (Arabian Sea, Bay of Bengal and Central India).

[Figure]

**Figure 2:** Scatter density plots of model-simulated albedo and the one computed using both statistical fitting method (nonlinear and multilinear fit) using satellite measurements for all three regions.

**First Indirect Radiative Forcing  (W m⁻²)**

[Figure]

[Figure]

[Figure]

[Figure]

[Figure]

**Figure 3.:** Comparison between satellite-based RFaci using both statistical fits and the one simulated by the SBDART model for all three regions and for all seasons. The different color indicates the regions, whereas the different symbols indicates the different seasons. Note that the fit is separately performed for each season and each region.

[Figure]

***Figure* 4.:** Seasonal variability of six-year averaged RF*aci* obtained using the nonlinear fit for all three regions for both anthropogenic and natural aerosols along with mean values.

**Table S4.:** The seasonal and regional variation of fitting parameters α1-α6 obtained from both multilinear and nonlinear fitting approaches.

| Area | Season | Nonlinear fit | | | | | | Multilinear regression fit | | | | | |
|------|--------|------|------|------|------|------|------|------|------|------|------|------|------|
| | | α1 | α2 | α3 | α4 | α5 | α6 | α1 | α2 | α3 | α4 | α5 | α6 |
| Arabian Sea | Winter | 0.158 | 0.023 | 0.376 | 0.232 | 0.199 | 0.943 | 0.098 | 0.101 | 0.002 | 0.005 | 1.000 | 0.303 |
| | Pre-Monsoon | 0.136 | 0.021 | -0.046 | 0.205 | 0.201 | 0.888 | 0.089 | 0.067 | 0.008 | 0.010 | 1.000 | 0.416 |
| | Monsoon | 0.109 | 0.029 | -0.046 | 0.395 | 0.201 | 0.888 | 0.092 | 0.049 | 0.009 | 0.012 | 1.000 | 0.422 |
| | Post-Monsoon | 0.154 | 0.026 | 0.108 | 0.010 | 1.192 | 0.172 | 0.091 | 0.097 | 0.044 | 0.024 | 1.000 | 0.558 |
| Bay of Bengal | Winter | 0.158 | 0.024 | -0.084 | 0.209 | 0.140 | 0.652 | 0.100 | 0.084 | 0.345 | 0.088 | 1.000 | 0.136 |
| | Pre-Monsoon | 0.127 | 0.012 | -0.043 | 0.081 | 0.081 | 0.474 | 0.092 | 0.060 | 0.004 | 0.002 | 1.000 | 0.324 |
| | Monsoon | 0.126 | 0.011 | -0.398 | 0.415 | 0.006 | 0.473 | 0.095 | 0.046 | 0.011 | 0.005 | 1.000 | 0.414 |
| | Post-Monsoon | 0.150 | 0.020 | 0.331 | 0.311 | 0.119 | 1.097 | 0.100 | 0.071 | 0.014 | 0.007 | 1.000 | 0.364 |
| Central India | Winter | 0.215 | 0.010 | -0.278 | 0.685 | 0.105 | 1.236 | 0.187 | 0.026 | 0.079 | 0.084 | 1.000 | 0.269 |
| | Pre-Monsoon | 0.183 | 0.003 | -0.259 | 0.662 | 0.099 | 1.339 | 0.171 | 0.018 | 0.020 | 0.009 | 1.000 | 0.278 |
| | Monsoon | 0.187 | 0.007 | 0.322 | 0.390 | 0.098 | 0.555 | 0.168 | 0.024 | 0.002 | 0.005 | 1.000 | 0.319 |
| | Post-Monsoon | 0.210 | 0.016 | -0.291 | 0.684 | 0.105 | 1.444 | 0.174 | 0.042 | 0.000 | 0.005 | 1.000 | 0.253 |

**Table S2.:** Mean seasonal variation of anthropogenic and natural fraction of aerosol optical depth over all three regions estimated using methodology by *Kim et al., (2007)*.

| | Arabian Sea (AS) | | Bay of Bengal (BOB) | | Central India (CI) | |
|---|---|---|---|---|---|---|
| | Anthro[*] | Nat[*] | Anthro | Nat | Anthro | Nat |
| Winter | 0.146±0.029 | 0.112±0.022 | 0.16±0.032 | 0.142±0.028 | 0.376±0.055 | 0.279±0.056 |
| Pre-Monsoon | 0.305±0.061 | 0.347±0.069 | 0.309±0.062 | 0.327±0.065 | 0.407±0.081 | 0.46±0.092 |
| Monsoon | 0.113±0.023 | 0.346±0.069 | 0.16±0.032 | 0.276±0.055 | 0.435±0.087 | 0.655±0.101 |
| Post-Monsoon | 0.309±0.062 | 0.298±0.059 | 0.305±0.061 | 0.234±0.046 | 0.627±0.095 | 0.505±0.089 |

[*]Anthro represents anthropogenic and Nat represents natural

[Figure]

**Table S3(a):** The statistics calculated for a different plausible values of surface albedo over land.

| Calculated (MODIS) Vs. Simulated (SBDART) | Land type | Land Surface Albedo | Nonlinear fit | | Multilinear fit | |
|---|---|---|---|---|---|---|
| | | | R | RMSE | R | RMSE |
| Planetary albedo | Present study | 0.15 | 0.74 | 0.017 | 0.65 | 0.065 |
| | Forest | 0.14 | 0.72 | 0.019 | 0.63 | 0.067 |
| | Cropland | 0.20 | 0.69 | 0.023 | 0.59 | 0.071 |
| | Grass land | 0.21 | 0.67 | 0.025 | 0.56 | 0.074 |
| | Barren land | 0.38 | 0.62 | 0.033 | 0.50 | 0.079 |
| First Indirect Forcing by Anthropogenic fraction | Present study | 0.15 | 0.83 | 0.037 | 0.62 | 0.048 |
| | Forest | 0.14 | 0.73 | 0.039 | 0.55 | 0.050 |
| | Cropland | 0.20 | 0.69 | 0.043 | 0.50 | 0.055 |
| | Grass land | 0.21 | 0.66 | 0.044 | 0.49 | 0.057 |
| | Barren land | 0.38 | 0.60 | 0.050 | 0.45 | 0.062 |
| First Indirect Forcing by Natural fraction | Present study | 0.15 | 0.77 | 0.042 | 0.54 | 0.049 |
| | Forest | 0.14 | 0.71 | 0.043 | 0.50 | 0.050 |
| | Cropland | 0.20 | 0.66 | 0.045 | 0.48 | 0.051 |
| | Grass land | 0.21 | 0.62 | 0.047 | 0.47 | 0.053 |
| | Barren land | 0.38 | 0.59 | 0.052 | 0.45 | 0.060 |

**Table S3(b):** The statistics calculated for the different plausible values of surface albedo over
ocean.

| Calculated (MODIS) Vs. Simulated (SBDART) | Ocean Surface Albedo | Nonlinear fit | | | | Multilinear fit | | | |
|---|---|---|---|---|---|---|---|---|---|
| | | R | | RMSE | | R | | RMSE | |
| | | AS | BOB | AS | BOB | AS | BOB | AS | BOB |
| Planetary albedo | Present study | 0.79 | 0.76 | 0.010 | 0.019 | 0.70 | 0.67 | 0.042 | 0.049 |
| | 0.13 | 0.69 | 0.67 | 0.021 | 0.029 | 0.61 | 0.59 | 0.059 | 0.059 |
| | 0.11 | 0.75 | 0.73 | 0.013 | 0.022 | 0.66 | 0.63 | 0.047 | 0.052 |
| | 0.08 | 0.71 | 0.7 | 0.017 | 0.026 | 0.63 | 0.61 | 0.053 | 0.055 |
| | 0.06 | 0.68 | 0.65 | 0.023 | 0.03 | 0.59 | 0.58 | 0.056 | 0.061 |
| First Indirect Forcing by Anthropogenic fraction | Present study | 0.89 | 0.87 | 0.035 | 0.037 | 0.68 | 0.68 | 0.047 | 0.048 |
| | 0.13 | 0.83 | 0.8 | 0.041 | 0.043 | 0.60 | 0.59 | 0.060 | 0.061 |
| | 0.11 | 0.87 | 0.85 | 0.037 | 0.04 | 0.64 | 0.64 | 0.055 | 0.054 |
| | 0.08 | 0.81 | 0.83 | 0.039 | 0.041 | 0.60 | 0.61 | 0.059 | 0.057 |
| | 0.06 | 0.78 | 0.78 | 0.042 | 0.045 | 0.58 | 0.56 | 0.065 | 0.067 |
| First Indirect Forcing by Natural fraction | Present study | 0.86 | 0.85 | 0.039 | 0.038 | 0.61 | 0.65 | 0.049 | 0.051 |
| | 0.13 | 0.74 | 0.76 | 0.046 | 0.047 | 0.51 | 0.57 | 0.055 | 0.060 |
| | 0.11 | 0.83 | 0.83 | 0.040 | 0.041 | 0.58 | 0.62 | 0.051 | 0.053 |
| | 0.08 | 0.80 | 0.79 | 0.042 | 0.044 | 0.54 | 0.60 | 0.054 | 0.057 |
| | 0.06 | 0.72 | 0.75 | 0.049 | 0.051 | 0.50 | 0.55 | 0.058 | 0.061 |

[Figure]

**Figure** **S1:** Scatter density plots CERES-retrieved α vs. calculated α for all three regions.